# A twin UGUA motif directs the balance between gene isoforms through CFIm and the mTORC1 signaling pathway

R Samuel Herron[1], Alexander K Kunisky[1], Jessica R Madden[1], Vivian I Anyaeche[1], May Z Maung[2], Hun-Way Hwang[1]*

[1]Department of Pathology, University of Pittsburgh, Pittsburgh, United States; [2]Department of Biological Sciences, University of Pittsburgh, Pittsburgh, United States

**Abstract** Alternative polyadenylation (APA) generates mRNA isoforms and diversifies gene expression. Here we report the discovery that the mTORC1 signaling pathway balances the expression of two *Trim9/TRIM9* isoforms through APA regulation in human and mouse. We showed that CFIm components, CPSF6 and NUDT21, promote the short *Trim9/TRIM9* isoform (*Trim9-S/TRIM9-S*) expression. In addition, we identified an evolutionarily conserved twin UGUA motif, UGUAYUGUA, in *TRIM9-S* polyadenylation site (PAS) that is critical for its regulation by CPSF6. We found additional CPSF6-regulated PASs with similar twin UGUA motifs in human and experimentally validated the twin UGUA motif functionality in *BMPR1B*, *MOB4*, and *BRD4-L*. Importantly, we showed that inserting a twin UGUA motif into a heterologous PAS was sufficient to confer regulation by CPSF6 and mTORC1. Our study reveals an evolutionarily conserved mechanism to regulate gene isoform expression by mTORC1 and implicates possible gene isoform imbalance in cancer and neurological disorders with mTORC1 pathway dysregulation.

*For correspondence:
hwhwang@pitt.edu

Competing interest: The authors declare that no competing interests exist.

## Editor's evaluation

This study presents a valuable finding on how mTORC signaling can impact metabolism by modulating the function of the APA machinery. The evidence supporting the claims of the authors is solid, with compelling data supporting the identification of a 'twin UGUA' motif that governs PAS selection by the CFIm complex, which is further connected to mTORC signaling. This work will have a general interest to those studying APA and cellular metabolism.

## Introduction

Proper expression of gene isoforms is important for normal physiology. For example, in neurons, co-expression of *Cdc42* exon 6 and exon 7 isoforms (both are alternative 3′ terminal exons) is crucial for normal morphogenesis as loss of either isoform results in abnormal development of axons and dendrites (*Yap et al., 2016*). However, for many genes, our knowledge about how their isoforms are regulated remains limited. Alternative polyadenylation (APA) is an important mechanism to generate RNA isoforms with different 3′ ends (*Gruber and Zavolan, 2019*; *Mitschka and Mayr, 2022*; *Tian and Manley, 2017*). The mRNA 3′-end processing factor CFIm is not essential for the cleavage reaction but has an outsized role in regulating APA as a sequence-dependent activator of mRNA 3′-end processing (*Boreikaite et al., 2022*; *Schmidt et al., 2022*; *Zhu et al., 2018*). The human CFIm consists of two subunits: a small subunit, CFIm25 (encoded by the NUDT21 gene), which directly binds to the UGUA motif, and two alternative large subunits, CFIm68 (encoded by the CPSF6 gene) and CFIm59

(encoded by the CPSF7 gene), which activate 3′-end processing by interacting with CPSF, one of the essential mRNA 3′-end processing factors (*Yang et al., 2011a*; *Zhu et al., 2018*). It has been shown that loss of NUDT21 or CPSF6, but not CPSF7, resulted in widespread and overlapping APA alterations in cells (*Ghosh et al., 2022*; *Gruber et al., 2012*; *Hwang et al., 2016*; *Li et al., 2015*; *Martin et al., 2012*; *Masamha et al., 2014*; *Zhu et al., 2018*). This is consistent with the finding that CFIm59 is a weaker activator of 3′-end processing compared with CFIm68 (*Zhu et al., 2018*) and highlights the important roles of NUDT21 and CPSF6 in APA regulation. Notably, APA changes from loss of NUDT21 or CPSF6 are predominantly proximal shifts that lead to 3′ UTR shortening or expression of truncated proteins (*Gruber et al., 2012*; *Martin et al., 2012*; *Masamha et al., 2014*), which can be explained by the skewed distribution of the UGUA motif that favors the distal PAS in CFIm target mRNAs (*Ghosh et al., 2022*; *Hwang et al., 2016*; *Li et al., 2015*; *Zhu et al., 2018*).

The mTORC1 signaling pathway plays a central role in regulating cell metabolism and hyperactive mTORC1 causes a group of neurodevelopmental disorders termed 'mTORopathies' with shared clinical manifestations (*Crino, 2016*; *Lipton and Sahin, 2014*; *Liu and Sabatini, 2020*). One of the best-studied mTORopathies is Tuberous Sclerosis Complex (TSC), which is caused by loss-of-function mutations in *TSC1* or *TSC2*, both of which are mTORC1 inhibitors (*Salussolia et al., 2019*). Interestingly, a recent RNA-seq study identified hundreds of mRNAs with shortened 3′ UTRs in *Tsc1*-null mouse embryonic fibroblasts (*Chang et al., 2015*), which is reminiscent of the aforementioned APA changes from loss of NUDT21 or CPSF6. Subsequently, a direct link between the mTORC1 pathway and CPSF6 was discovered in *Drosophila*—in starvation, repression of the mTORC1 signaling allows two downstream kinases, CDK8 and CLK2, to phosphorylate CPSF6, which is required for its nuclear localization to promote 3′ UTR lengthening of autophagy genes *Atg1* and *Atg8a* (*Tang et al., 2018*). The regulation of CPSF6 by CDK8 and CLK2 was also present in human MCF7 cells and was required for starvation-induced autophagy (*Tang et al., 2018*). Taken together, these two studies indicate that CPSF6-mediated APA regulation is a previously underappreciated component of the mTORC1 signaling pathway and suggest that APA dysregulation might contribute to TSC pathogenesis.

Our laboratory previously developed the cTag-PAPERCLIP technique, which utilizes a Cre-inducible allele of GFP-tagged poly(A)-binding protein (PABP) to perform high-throughput APA profiling in vivo in specific cell types without cell purification in mouse (*Hwang et al., 2017*). Because primary morbidity for TSC patients comes from central nervous system involvement (*Salussolia et al., 2019*), we sought to investigate how hyperactive mTORC1 impacts the APA landscape in different brain cell types in vivo using cTag-PAPERCLIP and the widely used *Tsc1* conditional knockout mice (*Bateup et al., 2013*; *Ercan et al., 2017*; *Kwiatkowski et al., 2002*; *Meikle et al., 2007*). During the investigation, we discovered that mTORC1 activities modulate the balance between two *Trim9/TRIM9* isoforms in mouse cortical excitatory neurons in vivo. *Trim9/TRIM9* encodes a neuronally enriched E3 ubiquitin ligase that regulates neuron morphogenesis (*Winkle et al., 2014*; *Winkle et al., 2016*), but it was not previously reported to be regulated by mTORC1 or CFIm. In this study, we present both the initial findings that lead to our discovery and the subsequent mechanistic studies characterizing the regulation of *Trim9/TRIM9* isoforms by mTORC1 through APA. We found that the expression of the short *Trim9/TRIM9* isoform was dependent on CPSF6 and NUDT21, and we identified an evolutionarily conserved twin UGUA motif (UGUAYUGUA) that is essential for its PAS usage. We further demonstrated the existence of similar functional twin-UGUA motifs in additional human CPSF6-dependent PASs. Importantly, we showed that it is possible to engineer a PAS to be regulated by CPSF6 and mTORC1 by insertion of a twin UGUA motif. Overall, our study identifies an evolutionarily conserved mechanism to regulate gene isoform expression by the mTORC1 pathway and expands current knowledge of APA regulation by CFIm beyond the UGUA motif.

## Results

### Systemic adeno-associated virus (AAV) delivery of Cre recombinase is an efficient way to model human disease in the cTag-PABP mouse for APA profiling

To conditionally knock out *Tsc1* in brain cortical excitatory neurons in cTag-PABP mice, we generated pAAV-Camk2a-iCre, an AAV vector that expresses codon-optimized Cre recombinase (iCre) under the excitatory neuron-specific mouse *Camk2a* promoter (*Madisen et al., 2010*), and injected adult

Tsc1-WT (Tsc1^{+/+}) and Tsc1-floxed (Tsc1^{fl/fl}) cTag-PABP mice with pAAV-Camk2a-iCre (**Figure 1A**). We sacrificed the injected mice 2–3 wk after injection and performed cTag-PAPERCLIP profiling to identify APA shifts resulting from activation of the mTORC1 signaling in brain cortical excitatory neurons in vivo (**Figure 1A**). We expected pAAV-Camk2a-iCre injection to (1) turn on PABP-GFP expression in excitatory neurons for cTag-PAPERCLIP profiling in both Tsc1-WT and Tsc1-floxed cTag-PABP mice and (2) knock Tsc1 out and activate the mTORC1 signaling in excitatory neurons in Tsc1-floxed cTag-PABP mice. To verify the expected effects from pAAV-Camk2a-iCre injection, we first examined Tsc1 expression in the cTag-PAPERCLIP profiles. As expected, Tsc1 mRNA expression was strongly reduced (>80%, 44.9 cpm to 8.6 cpm, p<0.01) in the injected Tsc1-floxed cTag-PABP mice (designated as Tsc1-KO hereafter) while both Tsc2 and Mtor mRNAs were expressed at similar levels in the injected mice of both genotypes (**Figure 1—figure supplement 1A**). Next, we checked whether the decrease in Tsc1 expression was sufficient to activate mTORC1 signaling in the Tsc1-KO cTag-PABP mice. We performed western blots for both total and phosphorylated S6 (PS6) ribosomal protein, a well-established indicator of mTORC1 activities (**Sengupta et al., 2010**). As expected, PS6 in the brain cortex was strongly increased in an injected Tsc1-floxed cTag-PABP mouse when compared to an uninjected mouse of the same genotype (**Figure 1—figure supplement 1B**). For comparison, we also examined the increase in PS6 in a genetically bred Camk2a-Cre; Tsc1^{fl/fl} mouse brain cortex (**Figure 1—figure supplement 1C**). We found that the magnitude of the PS6 increase is identical between AAV injection (**Figure 1—figure supplement 1B**) and genetic breeding (**Figure 1—figure supplement 1C**), indicating that Cre delivery efficiency from systemic AAV injection is equivalent to that of genetic breeding. Therefore, we concluded that systemic AAV injection is an effective way to deliver cell type-specific Cre recombinases for cTag-PAPERCLIP profiling.

## A shift toward *Trim9-L/TRIM9-L* expression in *Tsc1*-KO mouse brain and differentiated human *TSC2*-KO NSCs

From the cTag-PAPERCLIP experiments (two biological replicates for each genotype), we identified 135 genes with two PASs that significantly changed their APA preference (FDR < 0.05, >2-fold change) in Tsc1-KO cortical excitatory neurons—30 shifted proximally while 105 shifted distally (**Supplementary file 1**). APA shifts can be classified as UTR-APA (generating mRNA isoforms with different 3′ untranslated regions) or CDS-APA (generating mRNA isoforms with different coding sequences) (**Li et al., 2015**). We recently used a filtering strategy on PAPERCLIP data to identify a functional APA shift that protects cancer cells from chemotherapeutic agent-induced apoptosis (**Kunisky et al., 2021**). Therefore, to select APA shifts that are more likely to have functional impact for further investigation, we used the same strategy and searched for APA shifts that satisfied the following criteria: (1) switched the major PAS, defined as the predominantly used (>50% of total read counts) PAS between two PASs, and (2) CDS-APA. Only one gene from the list, Trim9, fulfilled both criteria. Trim9 encodes a neuronally enriched E3 ubiquitin ligase that regulates neuron morphogenesis (**Winkle et al., 2014**; **Winkle et al., 2016**). The two Trim9 isoforms identified by cTag-PAPERCLIP in mouse cortical excitatory neurons are designated as Trim9-L and Trim9-S hereafter. Tsc1 knockout in mouse cortical excitatory neurons causes a shift toward Trim9-L mRNA expression (**Figure 1B**). Trim9-L is the full-length isoform while Trim9-S uses an upstream PAS to skip exons 8–14 and lacks the SPRY domain at the protein level (**Figure 1B and C**). The SPRY domain is commonly found in TRIM family proteins and is implicated in protein–protein interaction (**Ozato et al., 2008**; **Perfetto et al., 2013**).

We next performed western blotting with brain cortices from Tsc1-WT and Tsc1-KO cTag-PABP mice to examine the impact of Trim9 APA shift at the protein level. In parallel with the cTag-PAPERCLIP results, a shift toward Trim9-L protein expression was also observed in the Tsc1-KO cTag-PABP mouse (**Figure 1D**). To exclude the possibility that the observed Trim9 APA shift is specific to pAAV-Camk2a-iCre-injected cTag-PABP mice, we measured expression of Trim9-L and Trim9-S mRNA isoforms by RT-qPCR using brain cortex tissues from the Camk2a-Cre; Tsc1^{fl/fl} mouse and Tsc1^{fl/fl} mouse pair shown in **Figure 1—figure supplement 1C**. Although this assay is not cell-type-specific like cTag-PAPERCLIP, a similar shift toward Trim9-L mRNA expression was also observed (**Figure 1E**). Lastly, because the Trim9 exon arrangement is conserved in human, we sought to determine whether activation of mTORC1 signaling also favors TRIM9-L expression in human cells of neuronal lineage. We measured the abundance of TRIM9-L and TRIM9-S isoforms in a published RNA-seq dataset generated from WT and TSC2-KO human neural stem cells (NSCs) after 6 wk of neural differentiation (**Grabole et al.,**

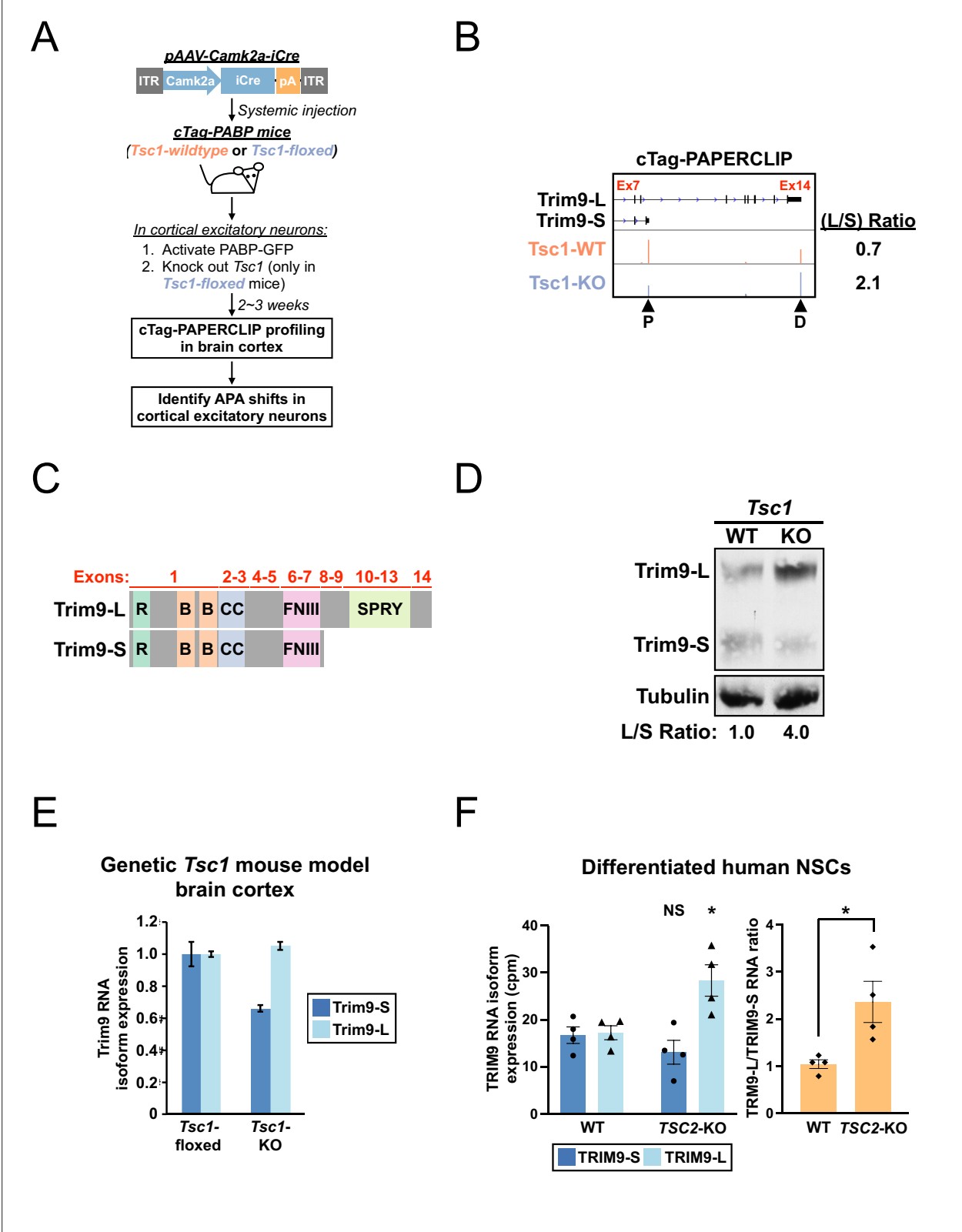

**Figure 1.** A shift toward *Trim9-L/TRIM9-L* expression in *Tsc1*-KO mouse cortical excitatory neurons and differentiated human *TSC2*-KO neural stem cells (NSCs). (**A**) The experimental strategy to identify in vivo alternative polyadenylation (APA) shifts in *Tsc1*-KO cortical excitatory neurons in mouse. (**B**) GENCODE annotations and cTag-PAPERCLIP results (merged from two biological replicates) for *Trim9*. Arrowheads: poly(A) sites identified by cTag-PAPERCLIP. P, proximal; D, distal. (Because *Trim9* is located on the minus strand, the orientation is horizontally flipped from the original.) (**C**) Illustrations

*Figure 1 continued on next page*

*Figure 1 continued*

comparing the exons and known protein domains (based on UniProt annotations) contained in mouse *Trim9-L and Trim9-S*. R, RING-type zinc finger. B, B box-type zinc finger; CC, coiled coil; FNIII, fibronectin type-III; SPRY, SPla/Ryanodine receptor. (**D**) Western blots showing a shift toward *Trim9-L* protein expression in a pAAV-Camk2a-iCre-injected *Tsc1^fl/fl^*; cTag-PABP mouse (KO) in the brain cortex when compared to an uninjected cTag-PABP mouse (WT). Tubulin: loading control. (**E**) Quantitation of *Trim9* mRNA isoforms by RT-qPCR in the brain cortex of *Tsc1^fl/fl^* ('*Tsc1*-floxed') and *Camk2a-Cre; Tsc1^fl/fl^* ('*Tsc1*-KO') mice (the same pair of mice shown in *Figure 1—figure supplement 1C*). Individual *Trim9* mRNA isoform expression was first normalized to *Rplp0* expression and then normalized to the expression level in the *Tsc1*-floxed mouse. (**F**) Quantitation of individual *TRIM9* mRNA isoforms (left) or *TRIM9-L/TRIM9-S* mRNA ratio (right) by RNA-seq (GSE78961) in *TSC2*-wildtype (WT) and *TSC2*-KO human NSCs after 6 wk of neural differentiation. cpm, counts per million. Error bars indicate SEM. Statistical significance is determined by two-tailed *t*-test (**F**, left panel) and one-tailed *t*-test (**F**, right panel). NS, not significant, *p<0.05.

The online version of this article includes the following source data and figure supplement(s) for figure 1:

**Source data 1.** *Figure 1D*, uncropped western blot images.

**Figure supplement 1.** Systemic adeno-associated virus (AAV) delivery of Cre recombinase activates mTORC1 signaling with similar efficiency to genetic breeding.

**Figure supplement 1—source data 1.** *Figure 1—figure supplement 1B*, uncropped western blot images.

**Figure supplement 1—source data 2.** *Figure 1—figure supplement 1C*, uncropped western blot images.

*2016*). We found that *TRIM9-L* mRNA expression was increased by more than 1.6-fold in *TSC2*-KO NSCs compared to WT NSCs (*Figure 1F*, left panel, 17.2 cpm to 28.4 cpm, p<0.05). Moreover, the *TRM9-L/TRIM9-S* ratio was consistently higher in *TSC2*-KO NSCs compared to WT NSCs in all replicates (*Figure 1F*, right panel). Taken together, these results show that, in both mouse and human neurons, hyperactive mTORC1 causes an APA shift in *Trim9/TRIM9* that favors full-length *Trim9-L/TRIM9-L* expression.

## The mTORC1 signaling pathway regulates *Trim9/TRIM9* isoform expression in mouse and human cells

We next sought to establish cell culture models to study the mechanistic link between the mTORC1 signaling pathway and *Trim9/TRIM9* APA in human and mouse. We first examined whether manipulation of mTORC1 activities in mouse Neuro-2a (N2a) cells would recapitulate the *Trim9* APA shift observed in mouse cortical neurons in vivo. Since mTORC1 is active under normal growth conditions in culture, we started by treating N2a cells with Torin 1, a potent mTORC1 inhibitor (*Thoreen et al., 2012*), and examined the expression of *Trim9-L* and *Trim9-S* by RT-qPCR and western blotting. As expected, Torin 1 treatment successfully shut down mTORC1 and eliminated PS6 in N2a cells (*Figure 2A*). We also performed siRNA experiments in N2a cells to characterize a TRIM9 antibody (*Qin et al., 2016*) for *Trim9-S* and *Trim9-L* protein detection (*Figure 2—figure supplement 1A*). As predicted from our in vivo findings (low mTORC1 activities would favor *Trim9-S* expression), in Torin 1-treated N2a cells, *Trim9* expression was consistently shifted toward *Trim9-S* with similar magnitudes at both mRNA (*Figure 2B*, right panel) and protein (*Figure 2C and D*) levels. Next, we went the opposite direction and evaluated whether we could further elevate mTORC1 activities in N2a cells by knocking down *Tsc1* or *Tsc2* using siRNAs (*Figure 2—figure supplement 1B*). As evidenced by a rise in PS6 (*Figure 2E*), we found that either siTsc1 or siTsc2 transfection could increase mTORC1 activities in N2a cells, which then favored *Trim9-L* mRNA expression as expected (*Figure 2F*, right panel).

Next, we evaluated human BE2C neuroblastoma cells, which can be differentiated into neuron-like cells and are a good host for siRNA transfection (*Ogorodnikov et al., 2018*). We measured *TRIM9-L* and *TRIM9-S* mRNA expression under three different conditions: control siRNA transfection (baseline), control siRNA transfection plus Torin 1 treatment (low mTORC1 activities), and *TSC2* siRNA transfection (high mTORC1 activities). As seen in mouse cortical neurons in vivo and in N2a cells, low mTORC1 activities indeed promoted *TRIM9-S* expression while high mTORC1 activities favored *TRIM9-L* expression in BE2C cells (*Figure 2G*, right panel, and *Figure 2—figure supplement 1C*). Taken together, our cell culture studies recapitulated the *Trim9* APA shift from *Tsc1*-KO mice and showed that the mTORC1 signaling pathway controls the balance between the two *Trim9/TRIM9* isoforms in both mouse and human.

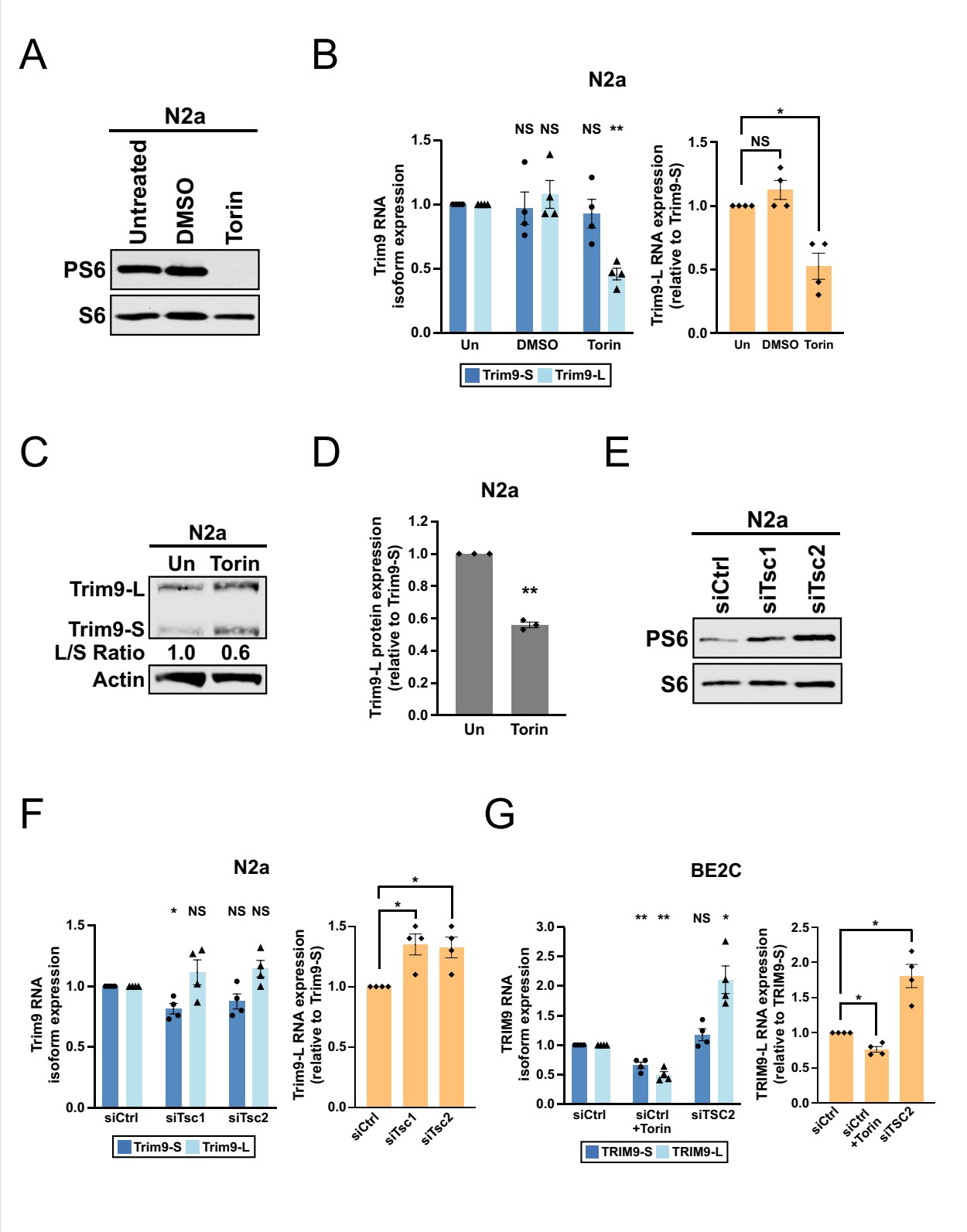

**Figure 2.** The mTORC1 signaling pathway regulates the balance between *Trim9-L/TRIM9-L* and *Trim9-S/TRIM9-S* in mouse and human cells. (**A**) Western blots showing the expression of total and phosphorylated S6 (PS6) ribosomal protein in N2a cells with different treatments for 48 hr. (**B**) Bar graphs showing the expression of two *Trim9* mRNA isoforms relative to the endogenous control (left panel) and the expression of *Trim9-L* mRNA relative to *Trim9-S* mRNA (right panel) measured by RT-qPCR in N2a cells receiving different treatments for 48 hr from four independent experiments (n = 4). In the

*Figure 2 continued on next page*

*Figure 2 continued*

left panel, individual *Trim9* mRNA isoform expression was first normalized to *Rplp0* expression and then normalized to the expression level in untreated cells (Un). In the right panel, *Trim9-L* expression was first normalized to *Trim9-S* expression and then normalized to the expression level in untreated cells. (**C**) Western blots showing the expression of *Trim9* protein isoforms in untreated (Un) or Torin 1-treated (for 48 hr) N2a cells. Actin: loading control. (**D**) Bar graphs showing the relative expression of *Trim9-L* protein to *Trim9-S* protein measured by western blotting in untreated (Un) or Torin 1-treated (for 48 hr) N2a cells from 3 independent experiments (n = 3). *Trim9-L* protein expression was first normalized to *Trim9-S* protein expression and then normalized to the expression level in untreated cells. (**E**) Western blots showing the expression of total and phosphorylated S6 ribosomal protein in N2a cells transfected with different siRNAs for 72 hr. siCtrl: control siRNA. (**F**) Bar graphs showing the expression of two *Trim9* mRNA isoforms relative to the endogenous control (left panel) and the expression of *Trim9-L* mRNA relative to *Trim9-S* mRNA (right panel) measured by RT-qPCR in N2a cells transfected with different siRNAs for 72 hr from four independent experiments (n = 4). (**G**) Bar graphs showing the expression of two *TRIM9* mRNA isoforms relative to the endogenous control (left panel) and the expression of *TRIM9-L* mRNA relative to *TRIM9-S* mRNA (right panel) measured by RT-qPCR in BE2C cells receiving different treatments (siCtrl+Torin: 48 hr; siCtrl and siTSC2: 72 hr) from four independent experiments (n = 4). In the left panel, individual *TRIM9* mRNA isoform expression was first normalized to *ACTB* expression and then normalized to the expression level in the siCtrl group. In the right panel, *TRIM9-L* expression was first normalized to *TRIM9-S* expression and then normalized to the expression level in the siCtrl group. Torin 1 was used at 250 nM in all experiments. Due to different exposure conditions, PS6 and S6 levels cannot be directly compared between (**A**) and (**E**). Error bars indicate SEM. Statistical significance is determined by two-tailed *t*-test. NS, not significant, *p<0.05; **p<0.01.

The online version of this article includes the following source data and figure supplement(s) for figure 2:

**Source data 1.** *Figure 2A*, uncropped western blot images.

**Source data 2.** *Figure 2C*, uncropped western blot images.

**Source data 3.** *Figure 2E*, uncropped western blot images.

**Figure supplement 1.** Characterization of a TRIM9 antibody and measurements of *Tsc1/Tsc2* siRNA knockdown efficiency.

**Figure supplement 1—source data 1.** *Figure 2—figure supplement 1A*, uncropped western blot images.

## CPSF6 and NUDT21 promote *Trim9-S/TRIM9-S* expression in mouse and human cells

Next, we wished to identify APA factor(s) that regulate *Trim9* APA. We first investigated a possible role of CFIm because mTORC1 signaling has been shown to modulate APA of autophagy genes through CPSF6 in *Drosophila* (**Tang et al., 2018**). We performed CRISPR gene editing to generate N2a cells with *Cpsf6* loss of function (**Figure 3A**). Interestingly, loss of *Cpsf6* had different effects on the two *Trim9* isoforms: it strongly decreased the abundance of *Trim9-S* mRNA but did not change *Trim9-L* mRNA expression (**Figure 3B**). Next, we induce *Nudt21* loss of function in N2a cells by siRNA transfection (**Figure 3C**), which phenocopied loss of *Cpsf6*: *Trim9-S* mRNA expression was decreased but *Trim9-L* mRNA expression did not statistically significantly change (**Figure 3D**). Lastly, we knocked down *CPSF6* and *NUDT21* separately by siRNAs in BE2C cells (**Figure 3E**). Both treatments similarly lowered *TRIM9-S* expression (**Figure 3F**). However, unlike in N2a cells, *CPSF6* knockdown also significantly increased *TRIM9-L* expression in BE2C cells (**Figure 3F**). Taken together, these results suggest that both CPSF6 and NUDT21 promote *Trim9-S/TRIM9-S* expression in mouse and human cells.

We next wished to address whether *Cpsf6* and *Nudt21* are required for the observed regulation of *Trim9* isoforms by the mTORC1 signaling pathway—high mTORC1 activities favor *Trim9-L* expression while low mTORC1 activities favor *Trim9-S* expression—using N2a cells. First, we inhibited mTORC1 in *Cpsf6* loss-of-function N2a cells and the corresponding control N2a cells (characterized in **Figure 3A and B**) with Torin 1. In control N2a cells, Torin 1 treatment shifted *Trim9* expression toward *Trim9-S* as expected (**Figure 3G** and **Figure 3—figure supplement 1A**). However, the shift was lost in *Cpsf6* loss-of-function N2a cells (**Figure 3G** and **Figure 3—figure supplement 1A**), suggesting a requirement of *Cpsf6*. Next, we examined how Torin 1 treatment affected *Trim9* expression in N2a cells with *Nudt21* loss of function from siRNA transfection (**Figure 3—figure supplement 1B and C**). In control siRNA-transfected N2a cells, Torin 1 treatment again shifted *Trim9* expression toward *Trim9-S* (**Figure 3H** and **Figure 3—figure supplement 1D**). In contrast, in *Nudt21* siRNA-transfected N2a cells, Torin 1 treatment did not favor *Trim9-S* expression but significantly shifted the balance toward *Trim9-L* instead (**Figure 3H** and **Figure 3—figure supplement 1D**). Altogether, these results support that *Cpsf6* and *Nudt21* are both required for the observed regulation of *Trim9* isoforms by the mTORC1 signaling pathway.

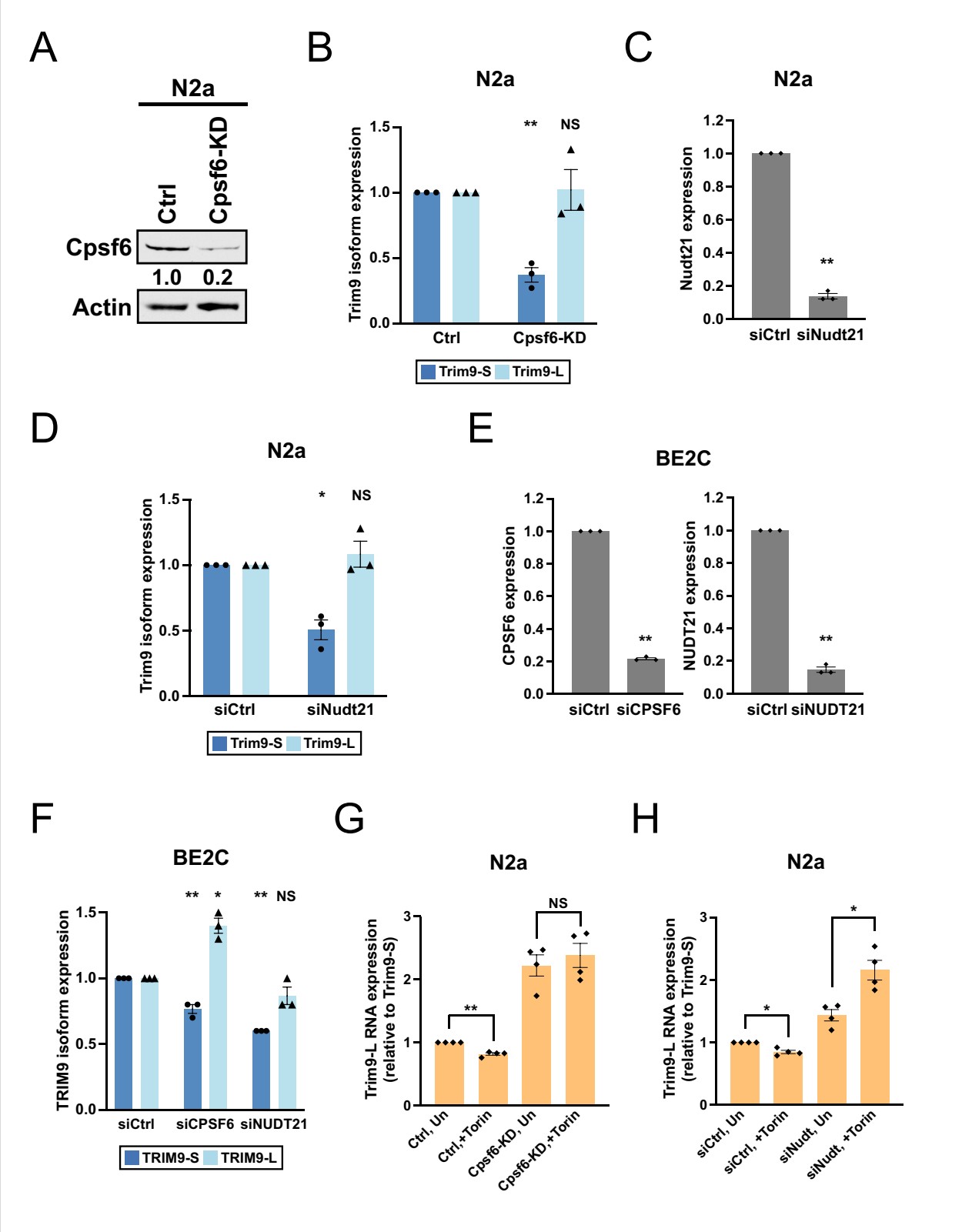

**Figure 3.** CPSF6 and NUDT21 promote *Trim9-S/TRIM9-S* expression in mouse and human cells. (**A**) Western blots showing *Cpsf6* protein expression in control (Ctrl) and *Cpsf6* knockdown (Cpsf6-KD) N2a cells. Actin: loading control. (**B**) Bar graphs showing the expression of *Trim9* mRNA isoforms measured by RT-qPCR in Ctrl and Cpsf6-KD N2a cells from three independent experiments (n = 3). (**C**) Bar graphs showing the knockdown efficiency of *Nudt21* siRNAs (siNudt21) measured 72 hr after transfection by RT-qPCR from three independent experiments (n = 3). siCtrl: control siRNA. (**D**) Bar

*Figure 3 continued on next page*

Figure 3 continued

graphs showing the expression of *Trim9* mRNA isoforms measured 72 hr after transfection by RT-qPCR in N2a cells from the same experiments in (**C**) (n = 3). (**E**) Bar graphs showing the knockdown efficiency of *CPSF6* (siCPSF6) and *NUDT21* (siNUDT21) siRNAs measured 72 hr after transfection by RT-qPCR from three independent experiments (n = 3). (**F**) Bar graphs showing the expression of *TRIM9* mRNA isoforms measured 72 hr after transfection by RT-qPCR in BE2C cells from the same experiments in (**E**) (n = 3). (**G**) Bar graphs showing the expression of *Trim9-L* mRNA relative to *Trim9-S* mRNA measured by RT-qPCR in Ctrl and Cpsf6-KD N2a cells from four independent experiments (n = 4). Ctrl and Cpsf6-KD N2a cells were either left untreated (Un) or treated with Torin for 48 hr (+Torin). (**H**) Bar graphs showing the expression of *Trim9-L* mRNA relative to *Trim9-S* mRNA measured by RT-qPCR in N2a cells receiving different treatments from four independent experiments (n = 4). See *Figure 3—figure supplement 1B* for the experiment design. Torin 1: 250 nM. Error bars indicate SEM. Statistical significance is determined by two-tailed *t*-test. NS, not significant, *p<0.05, **p<0.01.

The online version of this article includes the following source data and figure supplement(s) for figure 3:

**Source data 1.** *Figure 3A*, uncropped western blot images.

**Figure supplement 1.** Assaying *Cpsf6* and *Nudt21* requirement in *Trim9* regulation by mTORC1 activities in N2a cells.

## CPSF6 regulates *TRIM9-S* expression through an evolutionarily conserved twin UGUA motif

Because CFIm binds to the UGUA motif to promote mRNA 3′-end processing (*Yang et al., 2011a*; *Zhu et al., 2018*), we inspected the nucleotide sequence of mouse *Trim9-S* PAS and human *TRIM9-S* PAS for UGUA motifs. Both mouse *Trim9-S* PAS and human *TRIM9-S* PAS contain a 5′ twin UGUA motif (UGUAYUGUA; Y=C in human and T in mouse) followed by two downstream UGUA motifs, for a total of four UGUA motifs (*Figure 4A* and *Figure 4—figure supplement 1A*). Interestingly, three of the four UGUA motifs, including the twin UGUA motif, were conserved between mouse and human. We next wished to test the biological effects of the identified UGUA motifs on *TRIM9-S* PAS usage based on the well-established PAS competition assay (*Levitt et al., 1989*). In this assay, the PAS of interest were inserted into an expression construct upstream of a constant PAS, which provides the benchmark for comparing the usage of the PAS of interest with different mutations. In our tandem PAS reporter assay (*Figure 4B*), we used the bovine growth hormone (bGH) PAS, which does not contain any UGUA motif, as the reference PAS. Furthermore, we directly measured the abundance of mRNA isoforms generated from both PASs by RT-qPCR to infer relative usage of the PAS of interest instead of measuring protein products using a dual luciferase assay (*Lackford et al., 2014*).

We first validated the assay system using the L3 PAS, a direct CFIm target with two UGUA motifs 5′ to the poly(A) signal that are crucial for its usage (*Zhu et al., 2018*). We generated a wildtype L3 PAS reporter (L3-WT) (*Figure 4—figure supplement 1B*). To examine the *trans*-acting effects from CPSF6, we compared L3-WT usage between wildtype and *CPSF6*-knockout 293T cells (*Figure 4—figure supplement 1C*, referred to as CKO cells hereafter, *Sowd et al., 2016*). As expected, usage of L3-WT PAS was strongly decreased in CKO cells (*Figure 4C*). Next, to examine the *cis*-acting effects, we generated a mutant L3 PAS reporter (L3-MU) by mutating both UGUAs to UGGGs, a motif previously shown to abolish CFIm regulation (*Zhu et al., 2018*; *Figure 4—figure supplement 1B*). Consistent with the previous report (*Zhu et al., 2018*), usage of L3-MU was much lower compared to L3-WT in 293T cells (*Figure 4D*). Notably, the similar levels of decrease in L3 PAS usage between CPSF6 ablation (78%, *Figure 4C*) and mutations in both UGUA motifs (75%, *Figure 4D*) are consistent with the previous report that these two upstream UGUA motifs in L3 PAS are the main interaction sites with CFIm (*Zhu et al., 2018*). Altogether, these results demonstrated the sensitivity and validity of our tandem PAS reporter assay in detecting both *cis*- and *trans*-acting effects on PAS regulation.

Next, we proceeded to clone the human *TRIM9-S* PAS into our reporter (*Figure 4E*) and confirmed that the expected cleavage site was used by 3′ RACE (*Figure 4—figure supplement 1D*). To examine the contribution of UGUA motifs to PAS usage, we performed site-directed mutagenesis to generate a series of *TRIM9-S* PAS reporters with UGGG mutations introduced to different combinations of UGUA motifs (*Figure 4E*). Next, we performed reporter assays using all three *TRIM9-S* PAS reporters with the following results: (1) loss of CPSF6 approximately halved *TRIM9-S* PAS usage (*Figure 4F*, 51% decrease). (2) Mutations in the twin UGUA motif strongly decreased *TRIM9-S* PAS usage (*Figure 4G*, group 1 vs. group 2, 73% decrease). (3) Mutations in the other two UGUA motifs together had no effects on *TRIM9-S* PAS usage (*Figure 4G*, group 1 vs. group 3). (4) Loss of CPSF6 did not statistically significantly change the usage of *TRIM9-S* PAS with mutated twin UGUA motif (*Figure 4G*, group 2 vs. group 4, p=0.12). Altogether, these results support that CPSF6 promotes *TRIM9-S* PAS usage mostly

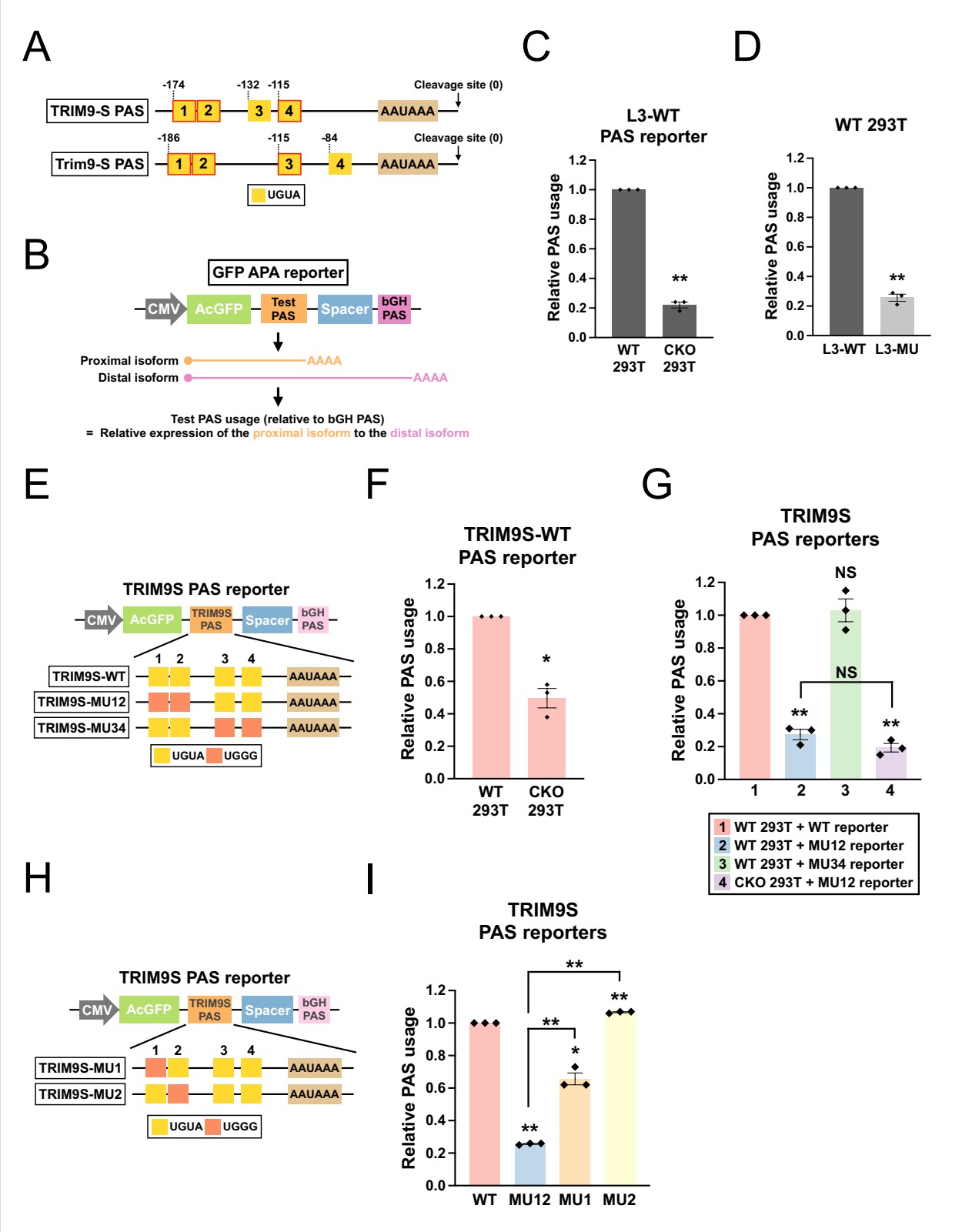

**Figure 4.** CPSF6 regulates *TRIM9-S* expression through an evolutionarily conserved twin UGUA motif. (**A**) Illustrations showing human *TRIM9-S* PAS and mouse *Trim9-S* PAS, including locations of the UGUA motifs (shown as yellow boxes numbered from 5′ to 3′). The twin UGUA motif is indicated by yellow boxes 1 and 2. The red outline indicates conservation between human and mouse. AAUAAA, the poly(A) signal. See *Figure 4—figure supplement 1A* for the alignment of human and mouse nucleotide sequences. (**B**) Illustrations showing the design of GFP alternative polyadenylation (APA) reporter

*Figure 4 continued on next page*

*Figure 4 continued*

to measure the strength of the inserted PAS (Test PAS) by RT-qPCR. See 'Materials and methods' for details. bGH, bovine growth hormone. (**C**) Bar graphs showing usage of L3 wildtype PAS in both wildtype (WT) and CPSF6-KO (CKO) 293T cells from three independent experiments (n = 3). (**D**) Bar graphs showing usage of both wildtype (L3-WT) and mutant (L3-MU) L3 PASs in 293T cells from three independent experiments (n = 3). (**E**) Illustrations showing the design of different *TRIM9S* PAS reporters. (**F**) Bar graphs showing usage of *TRIM9-S* wildtype PAS in both WT and CKO 293T cells from three independent experiments (n = 3). (**G**) Bar graphs showing usage of different *TRIM9S* PAS reporters in 293T or CKO cells from three independent experiments (n = 3). (**H**) Illustrations showing the design of additional *TRIM9S* PAS reporters. (**I**) Bar graphs showing usage of different *TRIM9S* PAS reporters in 293T cells from three independent experiments (n = 3). Different colors represent distinct PAS reporters. All measurements for the PAS reporter assays were performed 24 hr after transfection. Error bars indicate SEM. Statistical significance is determined by two-tailed $t$-test. NS, not significant, *p<0.05, **p<0.01.

The online version of this article includes the following source data and figure supplement(s) for figure 4:

**Figure supplement 1.** Characterization of reagents and tools for the PAS reporter assay.

**Figure supplement 1—source data 1.** *Figure 4—figure supplement 1C*, uncropped western blot images.

through the twin UGUA motif, which was evolutionarily conserved in multiple mammalian species including armadillo (*Figure 4—figure supplement 1E*).

To further examine the contribution from individual UGUAs of the twin UGUA motif to *TRIM9-S* PAS usage, we generated additional *TRIM9-S* PAS reporters (*Figure 4H*) and performed another series of reporter assays. Mutations in the first UGUA decreased *TRIM9-S* PAS usage but the effects were significantly weaker than those from mutations in both UGUAs (*Figure 4I*, MU1 vs. MU12, 34% vs. 75% decrease, p<0.01). In contrast, mutations in the second UGUA consistently but very modestly increased *TRIM9-S* PAS usage (*Figure 4I*, MU2: 7% increase). Taken together, these results suggest that, although the first UGUA of the *TRIM9-S* twin UGUA motif might play a larger role in PAS regulation than the second UGUA of the *TRIM9-S* twin UGUA motif, the effects from individual UGUAs of the twin UGUA motif are not additive and neither of them is solely responsible for the full effects from the entire twin UGUA motif.

## CPSF6-dependent PASs in *BMPR1B* and *MOB4* have functional twin UGUA motifs

Having identified a role of the *TRIM9-S* twin UGUA motif in the regulation of *TRIM9-S* PAS by CPSF6, we next asked whether we could find more CPSF6-regulated PASs that contain similar twin UGUA motifs in human. We generated BE2C cells that stably express an shRNA targeting CPSF6 from a tetracycline-inducible promoter (referred to as shCPSF6-BE2C cells hereafter). In shCPSF6-BE2C cells, 72 hr doxycycline treatment strongly reduced CPSF6 mRNA and protein expression as expected (*Figure 5—figure supplement 1A* and *Figure 5A*). Next, we performed APA profiling using PAPER-CLIP (*Hwang et al., 2016*) on shCPSF6-BE2C cells grown in the absence (high CPSF6) or presence (low CPSF6) of doxycycline to identify CPSF6-dependent PASs (*Figure 5B*; *Supplementary file 2*). We searched for 2-PAS genes that satisfied the following criteria: (1) switched the major PAS from loss of CPSF6 and (2) had a twin UGUA motif (UGUANUGUA) within 100 bp from the cleavage site in the loss-of-use PAS but not the other PAS. We identified six candidate genes—all of them have a twin UGUA motif in the distal PAS that lost use with low CPSF6 expression (*Supplementary file 3*). We chose the top 2 candidates, *BMPR1B* and *MOB4*, for experimental validation (*Figure 5C*). *BMPR1B* distal PAS has a third UGUA motif downstream of the UGUAUUGUA motif (*Figure 5D* and *Figure 5—figure supplement 1B*). In contrast, *MOB4* distal PAS does not have any additional UGUA motif besides the UGUACUGUA motif and a canonical poly(A) signal (*Figure 5D* and *Figure 5—figure supplement 1B*).

First, we performed RT-qPCR experiments measuring *BMPR1B* and *MOB4* mRNA isoforms in shCPSF6-BE2C cells to verify the loss of distal PAS usage identified by PAPERCLIP. We found that the distal mRNA isoform abundance indeed dropped in doxycycline-treated shCPSF6-BE2C cells for both genes (*Figure 5E*). Next, to examine the *cis*-regulatory effects of the identified twin UGUA motifs, we generated *BMPR1B* and *MOB4* distal PAS reporters with or without mutations in the twin UGUA motif (*Figure 5F*) for reporter assays in both 293T and CKO cells. The reporter assays showed that (1) loss of CPSF6 reduced usage of both *BMPR1B* and *MOB4* distal PASs (*Figure 5G and H*, group 1 vs. group 3, *BMPR1B*: 22% decrease, *MOB4*: 42% decrease); and (2) mutations in the twin UGUA motif diminished usage of both *BMPR1B* and *MOB4* distal PASs (*Figure 5G and H*, group 1 vs. group 2, *BMPR1B*: 19% decrease; *MOB4*: 63% decrease). These results confirm that CPSF6 promotes usage of both *BMPR1B*

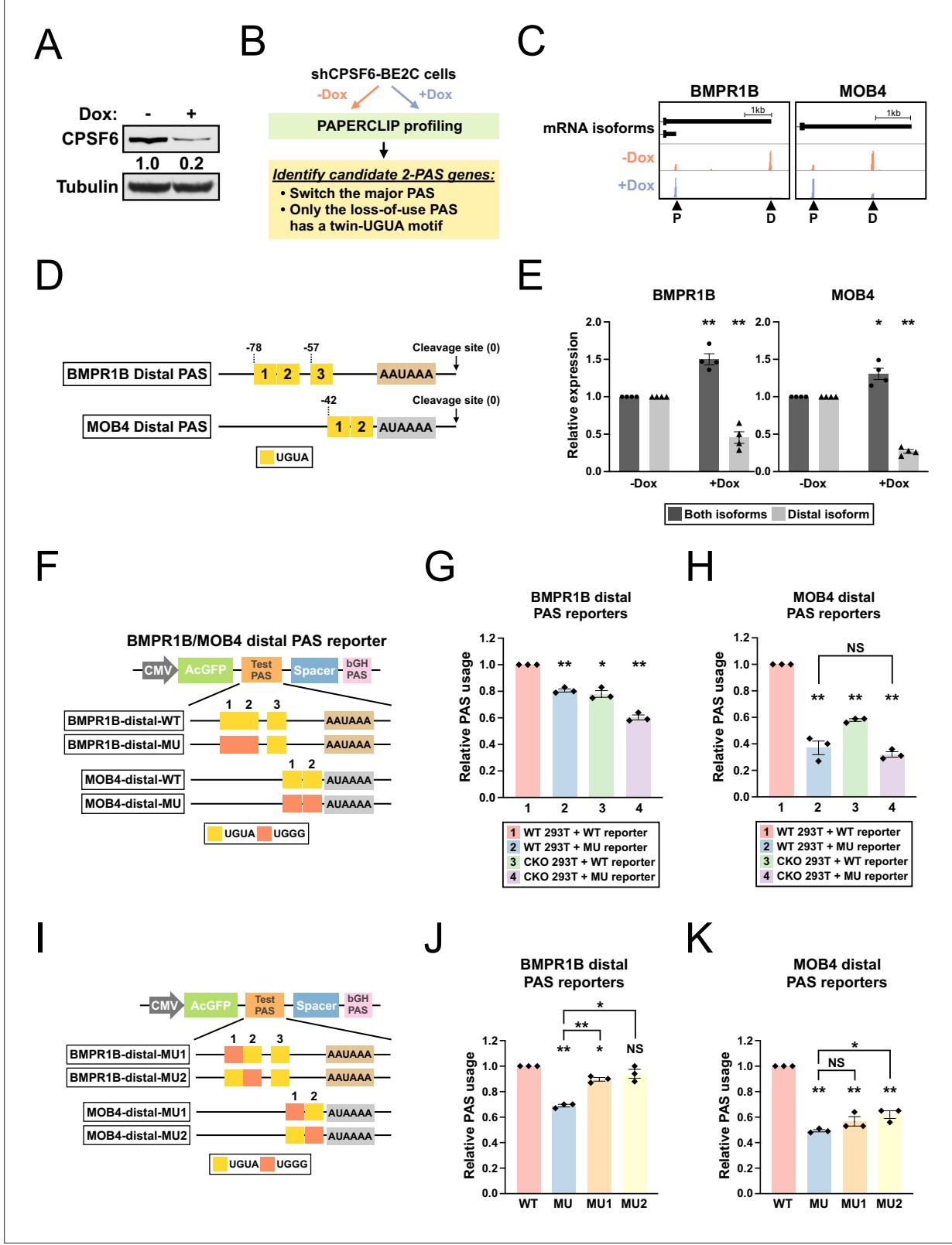

**Figure 5.** *BMPR1B* and *MOB4* distal PASs are two CPSF6-dependent PASs with a functional twin UGUA motif. (**A**) Western blots showing *CPSF6* protein expression in shCPSF6-BE2C cells with and without doxycycline treatment for 72 hr. Tubulin: loading control. (**B**) Illustrations showing the experimental strategy to identify candidate genes harboring a functional twin UGUA motif. (**C**) GENCODE annotations and PAPERCLIP results (merged from two biological replicates) in shCPSF6-BE2C cells for *BMPR1B* and *MOB4*. Arrowheads: poly(A) sites identified by PAPERCLIP. P, proximal; D,

*Figure 5 continued on next page*

*Figure 5 continued*

distal. (**D**) Illustrations showing human *BMPR1B* and *MOB4* distal PASs, including locations of the UGUA motifs (shown as yellow boxes numbered from 5' to 3'). The twin UGUA motif is indicated by yellow boxes 1 and 2. AAUAAA, the poly(A) signal. AUAAAA, a putative non-canonical poly(A) signal. See *Figure 5—figure supplement 1B* for the nucleotide sequences. (**E**) Bar graphs showing the expression of *BMPR1B* and *MOB4* mRNA isoforms measured by RT-qPCR in shCPSF6-BE2C cells with (+Dox) and without (-Dox) doxycycline treatment for 72 hr from four independent experiments (n = 4). (**F**) Illustrations showing the design of *BMPR1B* and *MOB4* distal PAS reporters. (**G, H**) Bar graphs showing usage of *BMPR1B* distal PAS (**G**) or *MOB4* distal PAS (**H**) in different combinations of 293T cells (WT and CKO) and reporters (WT and MU) from three independent experiments (n = 3). (**I**) Illustrations showing the design of additional *BMPR1B* and *MOB4* distal PAS reporters. (**J, K**) Bar graphs showing usage of different *BMPR1B* distal PAS reporters (**J**) or *MOB4* distal PAS reporters (**K**) in 293T cells from three independent experiments (n = 3). Different colors represent distinct PAS reporters. All measurements for the PAS reporter assays were performed 24 hr after transfection. Error bars indicate SEM. Statistical significance is determined by two-tailed *t*-test. NS, not significant, *p<0.05, **p<0.01.

The online version of this article includes the following source data and figure supplement(s) for figure 5:

**Source data 1.** *Figure 5A*, uncropped western blot images.

**Figure supplement 1.** Characterization of shCPSF6-BE2C cells and the genomic sequence of *BMPR1B* and *MOB4* distal PASs.

and *MOB4* distal PASs and demonstrates a functional role of the twin UGUA motif in both PASs. Importantly, loss of CPSF6 did not statistically significantly reduce the usage of *MOB4* distal PAS with the mutant twin UGUA motif (*Figure 5H*, group 2 vs. group 4, p=0.45), suggesting that regulation of *MOB4* distal PAS by CPSF6 occurs mostly through the twin UGUA motif just like in *TRIM9-S* PAS.

Lastly, we generated additional mutant *BMPR1B* and *MOB4* distal PAS reporters (*Figure 5I*) and performed another series of reporter assays to examine the contribution to PAS usage from individual UGUAs of the twin UGUA motifs in *BMPR1B* and *MOB4* distal PASs. In *BMPR1B* distal PAS, similar to *TRIM9-S* PAS, mutations in the first UGUA of the twin UGUA motif had weaker effects on PAS usage than those from mutations in both UGUAs (*Figure 5J*, MU1 vs. MU, 11% vs. 31% decrease, p<0.01) while mutations in the second UGUA of the twin UGUA motif did not statistically significantly decrease PAS usage (*Figure 5J*, MU2, 6% decrease, p=0.23). In contrast, in *MOB4* distal PAS, mutations in each UGUA of the twin UGUA motif drove the PAS usage down near the level caused by mutations in both UGUAs (*Figure 5K*, MU: 51% decrease; MU1: 43% decrease; MU2: 38% decrease).

## CPSF6 promotes expression of *BRD4-L*, which has a functional twin UGUA motif in its PAS

The bromodomain protein *BRD4* has two major isoforms, *BRD4-L* and *BRD4-S*, with opposing biological functions in MDA-MB-231 breast cancer cells (*Wu et al., 2020*). Although loss of CPSF6 in shCPSF6-BE2C cells did not cause a switch in the major PAS of *BRD4* (which remained *BRD4-S*), our PAPERCLIP profiling did identify *BRD4-L* PAS as one of the CPSF6-dependent PASs (*Figure 6A*). Moreover, although *BRD4-L* PAS is not conserved in mouse, we found that the sequence arrangement of *BRD4-L* PAS was similar to that of *TRIM9-S* PAS—both PASs have a 5' twin UGUA motif (UGUA-CUGUA) followed by two individual UGUA motifs at comparable downstream locations (*Figure 6B* and *Figure 6—figure supplement 1A*). Therefore, we sought to examine whether CPSF6 also regulates the balance between the two *BRD4* isoforms. We first measured *BRD4-L* and *BRD4-S* expression in 293T and CKO cells by RT-qPCR and western blots. In three independent experiments, we found that the *BRD4* mRNA and protein expression was consistently shifted toward *BRD4-S* in CKO cells when compared to 293T cells (*Figure 6—figure supplement 1B* and *Figure 6C–E*), suggesting that CPSF6 promotes *BRD4-L* expression.

We next examined whether CPSF6 also regulates *BRD4* isoform expression in MDA-MB-231 cells. We acutely knocked down CPSF6 in MDA-MB-231 cells with two different pools of CPSF6-targeting siRNAs (*Figure 6—figure supplement 1C*). Similar to the 293T/CKO experiments, we observed a consistent shift toward *BRD4-S* expression in siCPSF6-transfected MDA-MB-231 cells from three independent experiments (*Figure 6F* and *Figure 6—figure supplement 1D*). Moreover, the magnitude of decrease in the *BRD4-L* relative expression from loss of CPSF6 was similar between 293T cells and MDA-MB-231 cells (*Figure 6C and F*). To facilitate western blot analysis on BRD4 protein isoform expression, we generated shCPSF6-MDA-MB-231 cells that stably express an shRNA targeting CPSF6 from a tetracycline-inducible promoter (as in the shCPSF6-BE2C cells). As expected, in shCPSF6-MDA-MB-231 cells, doxycycline treatment strongly suppressed CPSF6 mRNA expression (*Figure 6—figure supplement 1E*) and again shifted the *BRD4* mRNA isoform balance towards *BRD4-S*

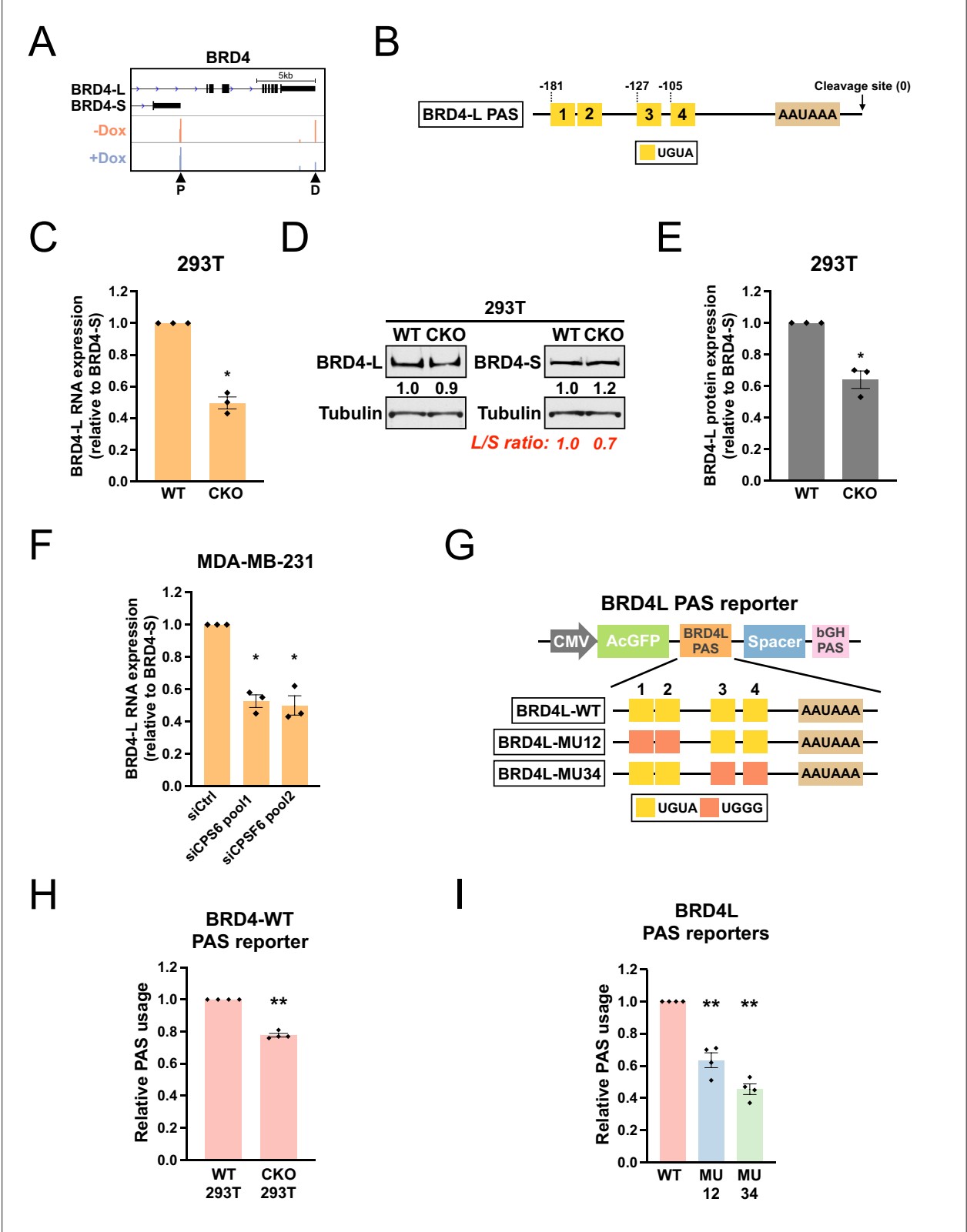

**Figure 6.** CPSF6 promotes expression of *BRD4-L*, which has a functional twin UGUA motif in the PAS. (**A**) GENCODE annotations and PAPERCLIP results (merged from two biological replicates) in shCPSF6-BE2C cells for *BRD4*. Arrowheads: poly(A) sites identified by PAPERCLIP. P, proximal; D, distal. (**B**) Illustrations showing human *BRD4-L* PAS, including locations of the UGUA motifs (shown as yellow boxes numbered from 5' to 3'). The twin UGUA motif is indicated by yellow boxes 1 and 2. AAUAAA, the poly(A) signal. See *Figure 6—figure supplement 1A* for the nucleotide sequence. (**C**) Bar

*Figure 6 continued on next page*

*Figure 6 continued*

graphs showing the expression of *BRD4-L* mRNA relative to *BRD4-S* mRNA measured by RT-qPCR in wildtype 293T (WT) and CKO cells from three independent experiments (n = 3). *BRD4-L* expression was first normalized to *BRD4-S* expression and then normalized to the expression level in WT. (**D**) Western blots showing the expression of *BRD4* protein isoforms in 293T and CKO cells. Tubulin: loading control. (**E**) Bar graphs showing the relative expression of *BRD4-L* protein to *BRD4-S* protein measured by western blotting in 293T and CKO cells from three independent experiments (n = 3). *BRD4-L* protein expression was first normalized to *BRD4-S* protein expression and then normalized to the expression level in WT. (**F**) Bar graphs showing the expression of *BRD4-L* mRNA relative to *BRD4-S* mRNA measured by RT-qPCR in MDA-MB-231 cells transfected with different siRNAs for 72 hr from three independent experiments (n = 3). siCtrl: control siRNA. *BRD4-L* expression was first normalized to *BRD4-S* expression and then normalized to the expression level in the siCtrl group. (**G**) Illustrations showing the design of *BRD4-L* PAS reporters. (**H**) Bar graphs showing usage of *BRD4-L* wildtype PAS in both 293T and CKO cells from four independent experiments (n = 4). (**I**) Bar graphs showing usage of different *BRD4-L* PAS reporters in 293T cells from four independent experiments (n = 4). Different colors represent distinct PAS reporters. All measurements for the PAS reporter assays were performed 24 hr after transfection. Error bars indicate SEM. Statistical significance is determined by two-tailed *t*-test. *p<0.05, **p<0.01.

The online version of this article includes the following source data and figure supplement(s) for figure 6:

**Source data 1.** *Figure 6D*, uncropped western blot images.

**Figure supplement 1.** *CPSF6* knockdown and *BRD4* isoform expression in MDA-MB-231 cells.

(*Figure 6—figure supplement 1F*). We next proceeded to examine *BRD4-L* and *BRD4-S* protein expression in shCPSF6-MDA-MB-231 cells with and without doxycycline treatment. Surprisingly, in contrast to the results from 293T/CKO cells, we did not find consistent expression changes between *BRD4-L* and *BRD4-S* proteins in response to doxycycline treatment. Nevertheless, these results from MDA-MB-231 and shCPSF6-MDA-MB-231 cells provide additional evidence to support that CPSF6 promotes *BRD4-L* mRNA expression.

Next, as we previously did with *TRIM9-S* PAS (*Figure 4E*), we generated a series of *BRD4-L* PAS reporters to examine the contribution of the UGUA motifs to *BRD4-L* PAS usage (*Figure 6G*). Consistent with the results from PAPERCLIP profiling, the WT *BRD4-L* PAS showed a consistent decrease in usage in CKO cells when compared to 293T cells (*Figure 6H*, 22% decrease). Furthermore, similar to *TRIM9-S* PAS, mutations in the twin UGUA motif indeed reduced *BRD4-L* PAS usage (*Figure 6I*, MU12: 36% decrease), supporting a role of the twin UGUA motif in promoting *BRD4-L* PAS usage. However, mutations in the two downstream UGUA motifs in *BRD4-L* PAS also reduced *BRD4-L* PAS usage (*Figure 6I*, MU34: 54% decrease), which was not the case for *TRIM9-S* PAS (*Figure 4G*). We concluded that, like *TRIM9-S* PAS, *BRD4-L* PAS is regulated by CPSF6 and has a functional twin UGUA motif. Nevertheless, *TRIM9-S* PAS and *BRD4-L* PAS are not regulated by CPSF6 in the same way despite similar UGUA motif arrangements. Importantly, we note that, through PAPERCLIP profiling and reporter assays in *BMPR1B*, *MOB4* and *BRD4-L*, we have expanded the number of human CPSF6-dependent PASs harboring a functional twin UGUA motif beyond *TRIM9-S*.

## Insertion of a twin UGUA motif into the *JUNB* PAS is sufficient to confer regulation by CPSF6 and mTORC1

We next wanted to test whether insertion of the *TRIM9-S* twin UGUA motif into a heterologous PAS is sufficient to make the host PAS responsive to CPSF6 and mTORC1 regulation in our reporter assay. For the host PAS, we selected human *JUNB* PAS as *JUNB* PAS does not contain any UGUA motif and its usage is not affected by loss of CPSF6 (*Hwang et al., 2016*; *Figure 7A* and *Figure 7—figure supplement 1A*). We found that insertion of the *TRIM9-S* twin UGUA motif strongly increased the *JUNB* PAS usage in 293T cells (*Figure 7B*, group 2, 84% increase) unlike the twin UGGG mutant (*Figure 7B*, group 5, 37% decrease). Importantly, this increase was entirely dependent on CPSF6 as it was completely abolished in CKO cells (*Figure 7B*, group 3). Interestingly, insertion of a single UGUA motif did not statistically significantly alter the *JUNB* PAS usage (*Figure 7B*, group 4, p=0.18), suggesting that a second copy of UGUA is essential for the biological activity of the *TRIM9-S* twin UGUA motif in this context.

Next, we sought to extend the *JUNB* PAS reporter assay to test whether the presence of the *TRIM9-S* twin UGUA motif sensitizes *JUNB* PAS to different levels of mTORC1 activities. Hyperactivation of mTORC1 was reported to inhibit CPSF6 function (*Tang et al., 2018*). Therefore, we hypothesized that mTORC1 hyperactivation would dampen usage of *JUNB*-2xUGUA PAS but not *JUNB*-WT PAS. Because HeLa cells offer a better model than 293T cells for mTORC1 hyperactivation (*Alesi et al., 2021*), we transfected HeLa cells with control siRNAs or *TSC1* siRNAs to hyperactivate mTORC1

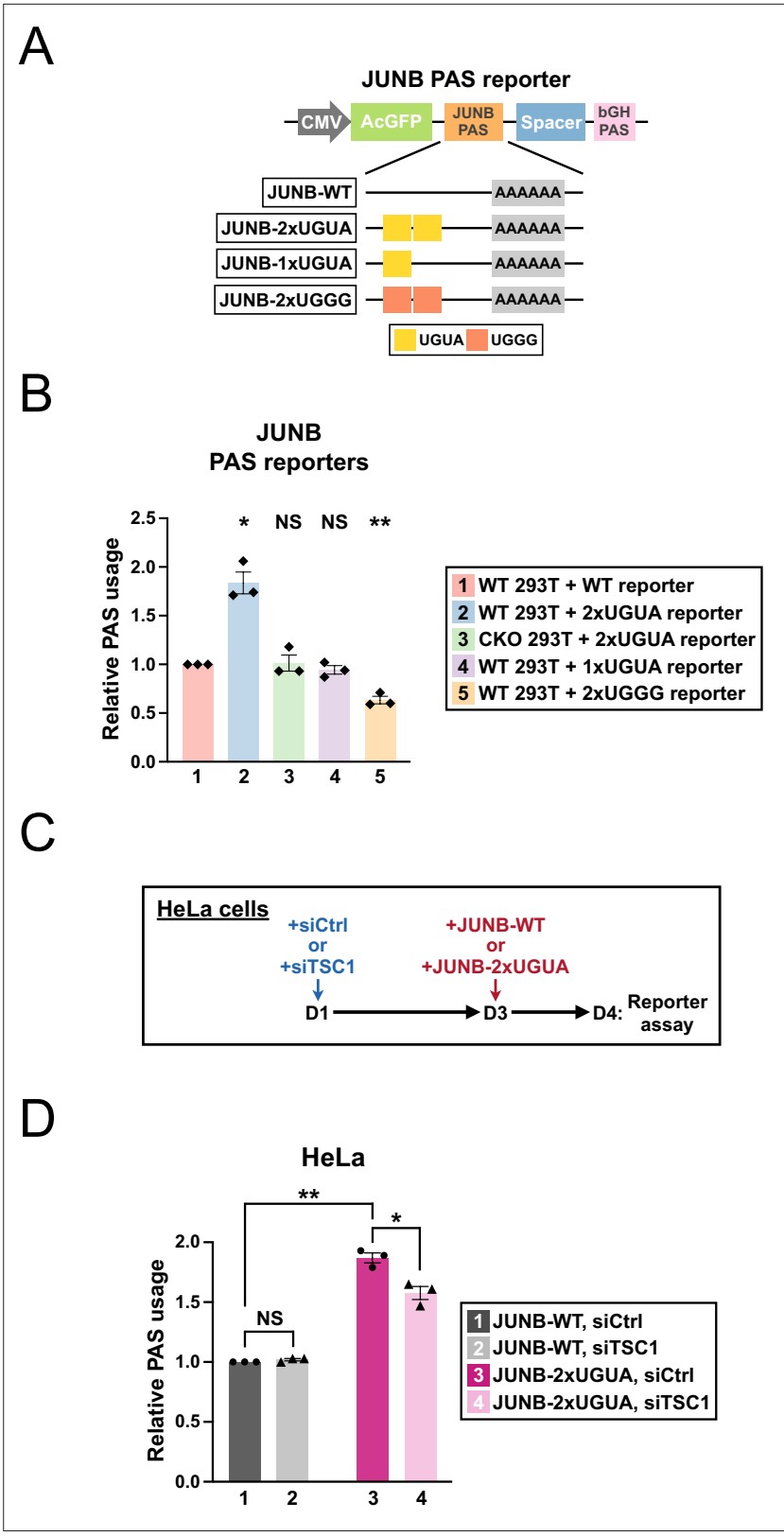

**Figure 7.** Insertion of a twin UGUA motif into the *JUNB* PAS is sufficient to confer regulation by CPSF6 and mTORC1. (**A**) Illustrations showing the design of *JUNB* PAS reporters. See *Figure 7—figure supplement 1A* for the nucleotide sequence. (**B**) Bar graphs showing usage of different *JUNB* PAS reporters (WT, 2xUGUA, 1xUGUA, and 2xUGGG) in 293T or CKO cells 24 hr after transfection from three independent experiments (n = 3).

*Figure 7 continued on next page*

*Figure 7 continued*

(**C**) Illustrations showing the experimental design for the *JUNB* PAS reporter assay in HeLa cells. (**D**) Bar graphs showing usage of *JUNB*-WT and *JUNB*-2xUGUA PAS reporters in HeLa cells with normal (siCtrl) or hyperactive (siTSC1) mTORC1 from three independent experiments (n = 3). Error bars indicate SEM. Statistical significance is determined by two-tailed *t*-test. NS, not significant, *p<0.05, **p<0.01.

The online version of this article includes the following figure supplement(s) for figure 7:

**Figure supplement 1.** Nucleotide sequences of wildtype and modified *JUNB* PASs for the reporter assay.

48 hr prior to a second transfection of either *JUNB*-WT or *JUNB*-2xUGUA reporters (*Figure 7C*). *TSC1* knockdown is indistinguishable between the two experimental groups of *JUNB*-WT and *JUNB*-2xUGUA (*Figure 7—figure supplement 1B*). As seen in 293T cells (*Figure 7B*), usage of *JUNB*-2xUGUA is strongly increased compared to that of *JUNB*-WT in HeLa cells transfected with control siRNAs (*Figure 7D*, group 1 vs. group 3, 87% increase). While usage of *JUNB*-WT was not affected by mTORC1 hyperactivation (*Figure 7D*, group 1 vs. group 2, p=0.18), usage of *JUNB*-2xUGUA was consistently decreased in siTSC1-transfected HeLa cells compared to siCtrl-transfected HeLa cells (*Figure 7D*, group 4 vs. group 3, 29% decrease). Taken together, these results demonstrate that insertion of the *TRIM9-S* twin UGUA motif into a heterologous PAS is sufficient to make the host PAS responsive to CPSF6 expression and mTORC1 activities.

## Discussion

The functional difference between *Trim9-L/TRIM9-L* and *Trim9-L/TRIM9-S* was reported in several biological contexts including neuron morphogenesis and glioblastoma progression (*Liu et al., 2018*; *Menon et al., 2015*; *Qin et al., 2016*; *Winkle et al., 2014*), but how the two *Trim9/TRIM9* isoforms are regulated remains unclear. In this study, we discovered that the balance between two *Trim9/TRIM9* isoforms was regulated by mTORC1 activities (*Figures 1 and 2*). We demonstrated that CPSF6 and NUDT21 are required for this regulation (*Figure 3*), and we further identified an evolutionarily conserved *cis*-regulatory motif, UGUAYUGUA, which plays a key role (*Figure 4* and *Figure 4—figure supplement 1*). In *Drosophila*, mTORC1 signaling was reported to regulate APA of autophagy genes, *Atg1* and *Atg8a*, through CPSF6 phosphorylation (*Figure 7—figure supplement 1C*; *Tang et al., 2018*). However, there are notable differences between the two regulatory events. First, *Atg1* and *Atg8a* PASs in *Drosophila* do not contain any twin-UGUA motif (*Figure 7—figure supplement 1D*). Second, the APA shifts in both *Atg1* and *Atg8a* lead to 3′ UTR lengthening and mRNA stabilization. In contrast, the APA shift in *Trim9/TRIM9* changes gene isoform expression—*Trim9-L/TRIM9-L* and *Trim9-S/TRIM9-S* proteins have distinct 3′ termini (*Figure 1C*) and different biological functions (see below). Therefore, although exactly how mTORC1 regulates CPSF6 to alter the balance between two *Trim9/TRIM9* isoforms in human and mouse remains to be further explored, our findings provide an example in which the mTORC1 signaling pathway alters protein isoform expression through APA regulation.

Our findings also expand current knowledge of APA regulation by CFIm beyond the UGUA motif. We provide multiple lines of evidence to support that the UGUAYUGUA motif is not distinct to *Trim9/TRIM9*. First, we have identified and experimentally confirmed the existence of a functional UGUA-YUGUA motif in additional human CPSF6-dependent PASs (*Supplementary file 3*). Second, despite the differences in the motif locations and the surrounding sequence contexts, the UGUAYUGUA motif plays a crucial role in CPSF6-mediated regulation in both *MOB4* distal PAS and *TRIM9-S* PAS (*Figures 4G and 5H*). Lastly, the UGUAYUGUA motif can be transferred to a heterologous PAS to gain CPSF6-mediated regulation (*Figure 7B*). Our findings indicate that the UGUAYUGUA motif is a naturally occurring *cis*-regulatory motif for CPSF6 in human and mouse that might be stronger than a single UGUA motif. Furthermore, our results suggest that, in order to fully grasp how CFIm regulates APA, it might be necessary to look beyond the simple UGUA motif and take different combinations of UGUA motifs into consideration as well.

Our reporter assay results suggest that it might be advantageous to have a second copy of UGUA: First, neither of the two UGUAs in the UGUAYUGUA motif is solely responsible for the full effects from the entire motif in *TRIM9-S* PAS (*Figure 4I*) and *BMPR1B* distal PAS (*Figure 5J*). Second, a single UGUA motif was insufficient to alter the *JUNB* PAS usage (*Figure 7B*). This is consistent with

a previous study reporting that the chance for NUDT21 binding is increased by having a second copy of UGUA in the RNA (*Yang et al., 2010*). The UGUAYUGUA motif is reminiscent of a 'bipartite motif,' which is a pair of short motifs spaced one or more bases apart and the type of motif preferred by some RNA-binding proteins with multiple RNA-binding domains (*Dominguez et al., 2018*; *Zhang and Darnell, 2011*). Although NUDT21 forms a dimer (each contains a Nudix domain for RNA binding) that can bind to two UGUA motifs simultaneously on the same RNA (*Yang et al., 2010*), the minimal distance between the two UGUA motifs that permits simultaneous binding is 7 nucleotides (*Yang et al., 2011b*). Therefore, it seems unlikely that the UGUAYUGUA motif functions by engaging both NUDT21 proteins in the dimer at the same time. Lastly, in human and mouse PASs, the UGUA motif is enriched in the 40–100 bp region upstream the cleavage site with a peak around 50 bp upstream the cleavage site (*Gruber and Zavolan, 2019*; *Wang et al., 2018*; *Zhu et al., 2018*). In contrast, the *Trim9-S/TRIM9-S* UGUAYUGUA motif is located more than 170 bp upstream of the cleavage site (*Figure 4A*). It will be interesting to understand how CFIm promotes 3'-end processing from this location in the future. It is worth noting that the mRNA 3'-end region is often folded, and forming secondary structures can promote 3'-end processing by shortening the actual distance to the cleavage site (*Wu and Bartel, 2017*).

Abnormal neuron morphogenesis has been reported in TSC (*Choi et al., 2008*; *Tavazoie et al., 2005*). In neurons, *Trim9-L* protein controls Netrin-mediated axon branching—it suppresses axon branching in the absence of Netrin-1 but permits it when Netrin-1 is present (*Winkle et al., 2014*). The response of *Trim9-L* protein to Netrin-1 depends on the SPRY domain, which interacts with the Netrin-1 receptor, DCC (*Winkle et al., 2014*). Because the *Trim9-S* protein lacks the SPRY domain, in *Trim9*-KO neurons rescued with *Trim9-S* protein the axon branching is constantly suppressed and unresponsive to Netrin-1 (*Menon et al., 2015*). Interestingly, in the mouse embryonic brain, *Trim9-S* protein seems to be the predominant isoform (*Winkle et al., 2014*; *Winkle et al., 2016*). If that is indeed the case, we speculate that a premature increase in *Trim9-L* protein due to mTORC1 hyperactivation in the susceptible developmental window might alter normal neuron morphogenesis and synapse formation. It is of great interest to elucidate the functional significance of *Trim9/TRIM9* isoform imbalance in TSC in future studies.

Taken together, our study reveals that mTORC1 hyperactivation causes *Trim9/TRIM9* isoform imbalance in TSC. Furthermore, we show that the UGUAYUGUA motif in the *Trim9-S/TRIM9-S* PAS and CFIm are key *cis*- and *trans*-acting factors linking mTORC1 to the balance between *Trim9/TRIM9* isoforms, respectively. Importantly, although transcriptional and translational abnormalities have been identified in *Tsc1*- and *Tsc2*-deficient neurons (*Dalal et al., 2021*; *Nie et al., 2015*), our results demonstrate that gene isoform imbalance is another mechanism for hyperactive mTORC1 to alter normal gene expression. As widespread dysregulation of gene isoform expression was recently found in cancer and neurological disorders (*Gandal et al., 2018*; *Kahles et al., 2018*), our results suggest that investigation of possible gene isoform imbalance in mTORopathies and cancer with hyperactive mTORC1 might provide important biological insights.

# Materials and methods

**Key resources table**

| Reagent type (species) or resource | Designation | Source or reference | Identifiers | Additional information |
|---|---|---|---|---|
| Genetic reagent (*Mus musculus*) | C57BL/6J | Jackson Laboratory | Cat#000664; RRID:IMSR_JAX:000664 | |
| Genetic reagent (*M. musculus*) | Tsc1tm1Djk/J | Jackson Laboratory | Cat#005680; RRID:IMSR_JAX:005680 | |
| Genetic reagent (*M. musculus*) | Camk2a-Cre | Jackson Laboratory | Cat#005359; RRID:IMSR_JAX:005359 | |
| Genetic reagent (*M. musculus*) | cTag-PABP | PMID:28910620 | MGI:6116824 | MGI symbol: Pabpc1tm1.2Rbd |
| Cell line (*M. musculus*) | Neuro-2a | ATCC | Cat#CCL-131; RRID:CVCL_0470 | |
| Cell line (*Homo sapiens*) | HEK293T/17 | ATCC | Cat#CRL-11268; RRID:CVCL_1926 | |
| Cell line (*H. sapiens*) | MDA-MB-231 | ATCC | Cat#HTB-26; RRID:CVCL_0062 | |
| Cell line (*H. sapiens*) | BE2C | ATCC | Cat#CRL-2268 | |

*Continued on next page*

*Continued*

| Reagent type (species) or resource | Designation | Source or reference | Identifiers | Additional information |
|---|---|---|---|---|
| Cell line (*H. sapiens*) | CKO HEK293T | PMID:26858452 | | |
| Cell line (*H. sapiens*) | shCPSF6-BE2C | This paper | | *Figure 5A* |
| Cell line (*H. sapiens*) | shCPSF6-MDA-MB-231 | This paper | | *Figure 6—figure supplement 1F* |
| Antibody | Anti-beta actin (mouse monoclonal) | ProteinTech | Cat#66009-1-Ig; RRID:AB_2687938 | 1:10000 |
| Antibody | Anti-alpha tubulin (mouse monoclonal) | Millipore | Cat#CP06; RRID:AB_2617116 | 1:3000 |
| Antibody | Anti-TRIM9 (rabbit polyclonal) | ProteinTech | Cat#10786-1-AP; RRID:AB_2209130 | 1:2000 |
| Antibody | Anti-S6, clone 5G10 (rabbit monoclonal) | Cell Signaling Technology | Cat#2217; RRID:AB_331355 | 1:2000 |
| Antibody | Anti-Phospho-S6 (Ser240/244) (rabbit monoclonal) | Cell Signaling Technology | Cat#5364; RRID:AB_10694233 | 1:2000 |
| Antibody | Anti-CPSF6 (rabbit polyclonal) | Bethyl Laboratories | Cat#A301-356A; RRID:AB_937781 | 1:2000 |
| Antibody | Anti-BRD4 (rabbit polyclonal) | Bethyl Laboratories | Cat#A301-985A; RRID:AB_1576498 | 1:10000 |
| Antibody | Anti-BRD4 (rabbit monoclonal) | Abcam | Cat#ab128874; RRID:AB_11145462 | 1:3000 |
| Strain, strain background | NEB 5-alpha | New England Biolabs | Cat#C2987 | |
| Strain, strain background | NEB Stable | New England Biolabs | Cat#C3040 | |
| Commercial assay or kit | ProtoScript II First Strand cDNA Synthesis Kit | New England Biolabs | Cat#E6560 | |
| Commercial assay or kit | Q5 Site-Directed Mutagenesis Kit | New England Biolabs | Cat#E0554 | |
| Commercial assay or kit | NEB PCR Cloning Kit | New England Biolabs | Cat#E1202 | |
| Commercial assay or kit | 10% Bis-Tris NuPAGE gels | Invitrogen | Cat#NP0301BOX | |
| Commercial assay or kit | 3~8% Tris-Acetate NuPAGE gels | Invitrogen | Cat#EA0375BOX | |
| Commercial assay or kit | DNase I | Invitrogen | Cat#18068015 | |
| Commercial assay or kit | Trizol reagent | Invitrogen | Cat#15596018 | |
| Commercial assay or kit | PerfeCTa SYBR Green SuperMix | QuantaBio | Cat#95054–500 | |
| Chemical compound, drug | DMSO | Sigma | Cat#D2650 | |
| Chemical compound, drug | Torin-1 | Cayman Chemical | Cat#10997 | |
| Chemical compound, drug | DharmaFECT1 | Horizon Discovery | Cat#T-2001-02 | |
| Chemical compound, drug | DharmaFECT2 | Horizon Discovery | Cat#T-2002-02 | |
| Chemical compound, drug | X-tremeGENE 9 | Sigma | Cat#XTG9-RO | |
| Sequence-based reagent | *Supplementary file 4* | This paper | | Primers used |
| Sequence-based reagent | *Supplementary file 5* | This paper | | siRNAs used |
| Recombinant DNA reagent | *Supplementary file 6* | Addgene or this paper | | Plasmids used or generated |
| Software, algorithm | CIMS package | PMID:24407355 | | |
| Software, algorithm | Kallisto | PMID:27043002 | | |
| Software, algorithm | Prism 9.5 | GraphPad Software | https://www.graphpad.com/; RRID:SCR_002798 | |

## Resource availability
### Materials availability
Plasmids and cell lines generated for this study will be shared by the corresponding author upon request.

## Experimental model and subject details

### Cell culture

N2a, HEK293T, CKO HEK293T, MDA-MB-231, and HeLa cells were grown in Dulbecco's modified Eagle's medium (DMEM). BE2C cells were grown in DMEM/F12. All media were supplemented with 10% FBS and penicillin-streptomycin. N2a, HEK293T, MDA-MB-231, and BE2C cells were obtained from ATCC with authentication and free of mycoplasma contamination. CKO HEK293T cells were provided by Dr. Alan Engelman. shCPSF6-BE2C and shCPSF6-MDA-MB-231 cells were generated by transducing wildtype BE2C and MDA-MB-231 cells with lentiviruses encoding a short hairpin RNA targeting CPSF6 (EZ-Tet-shCPSF6-Puro, see 'Method details' below) followed by puromycin selection. To generate Cpsf6-KD and Ctrl N2a cells, puromycin-resistant N2a cells that stably express Cas9 and a Cre-inducible blasticidin cassette were first transfected with either a plasmid encoding Cre and Cpsf6-targeting sgRNAs (Cpsf6-KD cells) or a plasmid encoding Cre but no sgRNAs (Ctrl cells) and subsequently selected with blasticidin and puromycin. SiRNA transfection was performed using DharmaFECT reagents (Horizon Discovery) with Silencer Select siRNAs (Invitrogen) or ON-TARGETplus siRNAs (Horizon Discovery) at the final concentration of 10 or 25 nM following the manufacturer's instructions. All siRNAs used are listed in *Supplementary file 5*. Plasmid transfection was performed using X-tremeGENE9 (MilliporeSigma) following the manufacturer's instructions.

Doxycycline was obtained from Sigma and was used at 1 μg/mL. Torin 1 were obtained from Cayman Chemical and was used at 250 nM.

### Mouse

All procedures were conducted according to the Institutional Animal Care and Use Committee (IACUC) guidelines at the University of Pittsburgh (Protocol #22102064). Camk2a-Cre and *Tsc1*-floxed mice (Tsc1$^{tm1Djk/J}$) were obtained from the Jackson Lab and were maintained as homozygotes. cTag-PABP mice were obtained from the Rockefeller University and were maintained by backcrossing to C57BL/6J. *Tsc1*-floxed cTag-PABP mice were generated by breeding *Tsc1*-floxed mice to cTag-PABP mice. For cTag-PAPERCLIP profiling, adult (8–12-week-old) *Tsc1*-wildtype or *Tsc1*-floxed cTag-PABP mice of both sexes received one-time retro-orbital injection with AAVs expressing iCre from mouse *Camk2a* promoter (pAAV-Camk2a-iCre) at the dose of 1 × 10$^{12}$ genome copies (gc), and they were housed for 2–3 wk before sacrifice. AAV was generated and packaged with the PHP.eB capsid, which broadly and efficiently transduces brain neurons from systemic AAV injection (*Chan et al., 2017*), by the University of Pennsylvania Vector Core. For *Tsc1*-floxed cTag-PABP mice, successful activation of mTORC1 signaling from injection was verified by S6 and Phosphor-S6 western blots before cTag-PAPERCLIP profiling. For *Figure 1—figure supplement 1C*, Camk2a-Cre; *Tsc1$^{fl/fl}$* mice were sacrificed at 4 wk of age because their survival decreased sharply afterward (*Bateup et al., 2013*).

## Method details

### cTag-PAPERCLIP, PAPERCLIP, and informatics analysis

cTag-PAPERCLIP and PAPERCLIP library construction was performed as previously described (*Hwang et al., 2017*; *Kunisky et al., 2021*). For cTag-PAPERCLIP, mouse brain cortices were crosslinked with 254 nm UV at 400 mJ/cm$^2$, lysed in 1× PXL buffer (1× PBS, 0.1% SDS, 0.5% NP-40, 0.5% sodium deoxycholate). For PAPERCLIP, cultured cells were crosslinked with 254 nm UV at 200 mJ/cm$^2$ and lysed in 1× TS buffer (1× PBS, 0.1% SDS, 1.0% Triton X-100). Both mouse brain and cell lysates were digested with DNase I (Promega) for 5 min at 37°C and then with RNase A (Thermo Fisher) for 5 min at 37°C. Lysates were cleared by centrifugation at 20,000 × *g* at 4°C for 10 min.

For cTag-PAPERCLIP, PABP-GFP-mRNA complexes were immunoprecipitated from cleared lysates for 2 hr at 4°C using Dynabeads protein G (Thermo Fisher) conjugated to anti-GFP (clones 19F7 and 19C8, MSKCC). Beads were then washed sequentially with 1× PXL buffer, 5× PXL buffer (5× PBS, 0.1% SDS, 0.5% NP-40, 0.5% sodium deoxycholate), and PNK buffer (50 mM Tris-HCl, pH 7.4, 10 mM MgCl$_2$, 0.5% NP-40). For PAPERCLIP, endogenous PABP-mRNA complexes were immunoprecipitated from cleared lysates for 2 hr at 4°C using Dynabeads protein G (Thermo Fisher) conjugated to anti-PABP (clone 10E10, Sigma). Beads were then washed sequentially with 1× TS buffer, 2× TS buffer (2× PBS, 0.1% SDS, 1.0% Triton X-100), and PNK buffer (50 mM Tris-HCl, pH 7.4, 10 mM MgCl$_2$, 0.5% NP-40). After wash, for both cTag-PAPERCLIP and PAPERCLIP, the immunoprecipitated protein-RNA complexes were treated with alkaline phosphatase and 5' labeled with $^{32}$P-gamma-ATP using T4

Polynucleotide Kinase on beads. The protein-RNA complexes were then eluted from beads, resolved on a Bis-Tris NuPAGE gel (8% for cTag-PAPERCLIP; 10% for PAPERCLIP), transferred to a nitrocellulose membrane, and film-imaged. Regions of interest were excised from the membrane and the RNA was isolated by Proteinase K digestion and phenol/chloroform extraction. Eluted RNA was reverse-transcribed using SuperScript III with BrdUTP. The resulting cDNAs were purified by two rounds of immunoprecipitation with Dynabeads protein G conjugated to anti-BrdU (clone IIB5, Millipore). The purified cDNAs were then ligated using CircLigase II (Lucigen) and PCR-amplified to generate the sequencing library.

Individual cTag-PAPERCLIP or PAPERCLIP libraries were multiplexed and sequenced by NextSeq (Illumina) to obtain 125-nt single-end reads. The procedures for raw read processing, mapping, and poly(A) site annotation were previously described (*Hwang et al., 2017*). Briefly, the raw reads were processed (filtered and collapsed) using the CIMS package (*Moore et al., 2014*) (now superseded by the CLIP tool kit [CTK]). Poly(A) sequence at the 3′ end was trimmed using CutAdapt (*Martin, 2011*). Trimmed reads that are longer than 25 nucleotides are aligned to mouse (mm10) or human (hg19) genome using Novoalign (http://www.novocraft.com/). The aligned reads were further processed using the CIMS package to remove PCR duplicates and cluster overlapping reads for poly(A) site identification.

For APA shift analysis, different cutoffs for the definition of two poly(A) site genes were computed to generate a broad set of two poly(A) site genes for comparison. Statistical analysis for APA shift was previously described (*Hwang et al., 2016*) and is described below: EdgeR package (*Robinson et al., 2010*) was used to statistically test significant APA shifts between two experimental conditions, while accounting for biological and technical variability between experimental replicates. Each replicate dataset was first normalized to account for library size and compositional bias using the TMM methodology (*Robinson and Oshlack, 2010*). The poly(A) site PAPERCLIP read count data were modeled as a negative binomial distribution and fitted a generalized linear model (GLM) with explanatory variables for batch, experimental, poly(A) location, and an interaction factor for experimental condition × poly(A) location. For each gene, GLM likelihood ratio test was conducted to test if the interaction coefficient between experimental condition and poly(A) location was non-zero. GLM likelihood ratio test-derived p-values were adjusted for multiple hypotheses testing using the qvalue package (*Storey and Tibshirani, 2003*). Significant APA shift is defined as FDR < 0.05 and a >2-fold change of (proximal PAS/distal PAS) ratio between experimental conditions. All gene lists are provided (*Supplementary files 1 and 2*). Kallisto (*Bray et al., 2016*) was used to estimate TRIM9 isoform abundance in GSE78961 (*Figure 1F*).

## SDS-PAGE and western blots

20–60 µg lysates from culture cells or mouse tissues were separated on 10% Bis-Tris or 3~8% Tris-Acetate Novex NuPAGE gels (Invitrogen) and transferred to nitrocellulose membrane following standard procedures. The following antibodies are used for western blotting: mouse monoclonal anti-beta actin (ProteinTech, 66009-1-Ig), mouse monoclonal anti-alpha tubulin (Millipore, CP06), mouse monoclonal anti-HA (clone 16B12, BioLegend, 901501), rabbit polyclonal anti-TRIM9 (ProteinTech, 10786-1-AP), rabbit monoclonal anti-S6 ribosomal protein (Cell Signaling Technology, 2217), rabbit monoclonal anti-Phospho-S6 ribosomal protein (Ser240/244) (Cell Signaling Technology, 5364), rabbit polyclonal anti-CPSF6 (Bethyl Labs, A301-356A), rabbit polyclonal anti-BRD4 (Bethyl Labs, A301-985A), and rabbit monoclonal anti-BRD4 (Abcam, ab128874).

## Reverse transcription and quantitative PCR (RT-qPCR)

qPCR was performed using PerfeCTa SYBR Green SuperMix (QuantaBio) in triplicates. All primer sequences are listed in *Supplementary file 4*. For mRNA quantification, reverse transcription was performed using ProtoScript II First Strand cDNA Synthesis Kit (NEB) using d(T)$_{23}$VN primer with DNase I (Invitrogen) digestion on 1 µg total RNA generated from Trizol (Invitrogen) extraction. The cycling parameters for qPCR were 95°C for 10 min. followed by 40 cycles of 95°C for 15 s, 58°C for 30 s, and 72°C for 20 s. Quantification was calculated using the ΔΔCt method with the following endogenous controls: ACTB (human) and Rplp0 (mouse).

For PAS reporter assay, reverse transcription was performed using ProtoScript II First Strand cDNA Synthesis Kit (New England Biolabs) using an anchored d(T) primer (R1-T25-VN) with DNase I

(Invitrogen) digestion on 1 µg total RNA generated from Trizol (Invitrogen) extraction. qPCR quantification of the proximal and distal mRNA isoforms generated from each PAS reporter is performed using a PAS specific forward primer and a common reverse primer complementary to the anchoring sequence of R1-T25-VN. The cycling parameters for qPCR were 95°C for 10 min, followed by 40 cycles of 95°C for 15 s, 58°C for 30 s, and 72°C for 8 s. Quantification was calculated using the ΔΔCt method relative to the distal mRNA isoform that uses the bGH PAS.

## Cloning and constructs

Standard cloning procedure (restriction digest, ligation, and transformation) was performed to generate the desired constructs. All insert sequences were verified by Sanger sequencing. Oligonucleotides and primers are listed in *Supplementary file 4*. pAAV-Camk2a-iCre was generated by replacing GFP in pAAV-CAMKII-GFP (Addgene, 64545) with iCre (a codon-optimized Cre recombinase), which was amplified from pEMS1985 (Addgene, 49116) by PCR. EZ-Tet-shCPSF6-Puro was generated by inserting a short hairpin RNA targeting CPSF6 into EZ-Tet-pLKO-Puro (Addgene, 85966) between the NheI and EcoRI sites. GFP APA reporter was generated through the following modifications of the pcDNA 3.1 plasmid: (1) insertion of AcGFP between the NheI and XbaI sites, and (2) insertion of a spacer sequence in front of the bGH poly(A) signal using the BamHI site. The following PAS reporters were generated by insertion of oligonucleotides into the GFP APA reporter between the XhoI and XbaI sites: L3-WT, L3-MU, BMPR1B-distal-WT, BMPR1B-distal-MU, BMPR1B-distal-MU1, BMPR1B-distal-MU2, MOB4-distal-WT, MOB4-distal-MU, MOB4-distal-MU1, MOB4-distal-MU2, JUNB-WT, JUNB-2xUGUA, JUNB-1xUGUA, JUNB-2xUGGG. TRIM9S-WT, and BRD4L-WT PAS reporters were generated by inserting TRIM9-S and BRD4-L PASs (amplified from human genomic DNA) into the GFP APA reporter between the XhoI and XbaI sites. TRIM9S-MU12, TRIM9S-MU34, TRIM9S-MU1, TRIM9S-MU2, BRD4L-MU12, and BRD4L-MU34 PAS reporters were generated by site-directed mutagenesis of TRIM9S-WT or BRD4L-WT PAS reporters using Q5 Site-Directed Mutagenesis Kit (New England Biolabs).

## Quantification and statistical analysis

Details of the statistical tests are indicated below and in the figure legends. Statistical analyses were performed using R.

For *Figures 1F (left panel), 2B, D, F,G,3B–H, 4C, D, F, G,I, 5E, G, H,J, K,6C,E, F, H, I, 7B, D*, *Figure 1—figure supplement 1A*, *Figure 2—figure supplement 1B and C*, *Figure 3—figure supplement 1A*, *Figure 3—figure supplement 1C and D*, *Figure 5—figure supplement 1A*, *Figure 6—figure supplement 1B–F*, and *Figure 7—figure supplement 1B*, statistical significance was determined by two-tailed Welch two-sample *t*-test.

For *Figure 1F* (right panel), statistical significance was determined by one-tailed Welch two-sample *t*-test.

For all figures: *p<0.05, **p<0.01.

## Acknowledgements

We thank Dr. Alan Engelman for providing CKO HEK293T cells. We also thank the Division of Laboratory Animal Resources (DLAR) at the University of Pittsburgh for technical assistance in mouse injection and the Vector Core at the University of Pennsylvania for AAV production. We acknowledge the Health Sciences Sequencing Core at UPMC Children's Hospital of Pittsburgh, the University of Pittsburgh Center for Research Computing, and the University of Pittsburgh HSCRF Genomics Research Core for high-throughput sequencing and Sanger sequencing. This work was supported by a grant from the National Institutes of Health (NS113861 to H-WH).

## Additional information

### Funding

| Funder | Grant reference number | Author |
|---|---|---|
| National Institutes of Health | NS113861 | Hun-Way Hwang |

The funders had no role in study design, data collection and interpretation, or the decision to submit the work for publication.

### Author contributions

R Samuel Herron, Formal analysis, Investigation, Methodology, Writing – review and editing; Alexander K Kunisky, Jessica R Madden, Vivian I Anyaeche, May Z Maung, Investigation; Hun-Way Hwang, Conceptualization, Formal analysis, Supervision, Funding acquisition, Investigation, Methodology, Writing - original draft, Writing – review and editing

### Author ORCIDs

R Samuel Herron ⓘ http://orcid.org/0000-0003-0212-9070
Alexander K Kunisky ⓘ http://orcid.org/0000-0002-6974-7474
Jessica R Madden ⓘ http://orcid.org/0009-0003-4046-1733
Vivian I Anyaeche ⓘ http://orcid.org/0000-0002-8912-539X
Hun-Way Hwang ⓘ http://orcid.org/0000-0003-0522-1577

### Ethics

All procedures were conducted according to the Institutional Animal Care and Use Committee (IACUC) guidelines at the University of Pittsburgh (Protocol #22102064).

### Decision letter and Author response

Decision letter https://doi.org/10.7554/eLife.85036.sa1
Author response https://doi.org/10.7554/eLife.85036.sa2

## Additional files

### Supplementary files

- Supplementary file 1. Gene lists from cTag-PAPERCLIP profiling in *Tsc1*-WT/KO mice.
- Supplementary file 2. Gene lists from PAPERCLIP profiling in shCPSF6-BE2C cells.
- Supplementary file 3. Identified human genes with twin-UGUA motif-containing PAS.
- Supplementary file 4. List of oligonucleotides and primers.
- Supplementary file 5. List of siRNAs.
- Supplementary file 6. List of plasmids.
- MDAR checklist

### Data availability

cTag-PAPERCLIP and PAPERCLIP data have been deposited at GEO under accession code GSE210768.

The following dataset was generated:

| Author(s) | Year | Dataset title | Dataset URL | Database and Identifier |
|---|---|---|---|---|
| Herron RS, Kunisky AK, Madden JR, Hwang HW, Anyaeche VI, Maung MZ | 2022 | A twin UGUA motif directs the balance between gene isoforms through CFIm and the mTORC1 signaling pathway | https://www.ncbi.nlm.nih.gov/geo/query/acc.cgi?acc=GSE210768 | NCBI Gene Expression Omnibus, GSE210768 |

The following previously published dataset was used:

| Author(s) | Year | Dataset title | Dataset URL | Database and Identifier |
|---|---|---|---|---|
| Grabole N, Zhang JD, Aigner S, Ruderisch N | 2016 | Modeling the Neuropathology of Tuberous Sclerosis with Human Stem Cells Reveals a Role for Inflammation and Angiogenic Growth Factors | https://www.ncbi.nlm.nih.gov/geo/query/acc.cgi?acc=gse78961 | NCBI Gene Expression Omnibus, GSE78961 |

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
