## [Editor Report]

This study presents a valuable finding on how mTORC signaling can impact metabolism by modulating the function of the APA machinery. The evidence supporting the claims of the authors is solid, with compelling data supporting the identification of a 'twin UGUA' motif that governs PAS selection by the CFIm complex, which is further connected to mTORC signaling. This work will have a general interest to those studying APA and cellular metabolism.

---

## [Decision Letter]

**Decision letter after peer review:**

Thank you for submitting your article "A twin UGUA motif directs the balance between gene isoforms through CFIm and the mTORC1 signaling pathway" for consideration by *eLife*. Your article has been reviewed by 3 peer reviewers, one of whom is a member of our Board of Reviewing Editors, and the evaluation has been overseen by James Manley as the Senior Editor. The reviewers have opted to remain anonymous.

Essential revisions:

1) We encourage the authors to consider addressing many of the issues raised by the Reviewers and we note that several of the issues can be addressed textually while others can with minimal additional experimentation. One specific experiment that was a consensus concern of the Reviewers is: Showing that knockdown of CPSF6/NUDT21 makes TRIM9 polyadenylation insensitive to mTORC1 dysregulation (e.g. with Torin or Tsc1 KD), to demonstrate whether the change in TRIM9 isoform ratio seen during mTORC1 dysregulation directly requires CFIm and could not occur through an independent pathway.

*Reviewer #1 (Recommendations for the authors):*

1. The initial descriptions of the genetic challenges of the breeding schemes of the mice used is almost impenetrable. Much of this could be shifted to the methods. I appreciate the authors attention to technical detail, but momentum is immediately lost in this paper without much gain in technical resolution. The authors should consider consolidating the language.

2. The description of the two criteria for selecting candidate APA events for further investigation included 'resulted in expression of two different protein isoforms'. The authors need to clarify and justify this criteria. Only ~10% of APA events are splicing-APA in that distinct C-termini are created. By implementing this criteria, the authors artificially trimmed the list down by a huge margin. I think this is still ok, but more explanation is required so that the average reader understands what went into this filtering event.

3. Can the authors describe what a SPRY domain is for non-experts?

4. The section of the results focused on cdk8 or CLK2 is under-developed. What doe cdk8 or CLK2 inhibitors do in the absence of Torin1? Can the authors show that cdk8 inhibition does anything to phosphorylation of CPSF6? As it stands, this section is not well-controlled or analyzed and needs improvement.

5. Continuing along the lines of the previous point, CPSF6 phosphorylation is also under-developed. Where are these phospho-mutants localized? Do cdk8 manipulations also effect these reporter experiments? While the UGUA mutagenesis experiments are well done, the authors could consider removing the phosphorylation experiments altogether and not have their overall thesis impacted at all.

*Reviewer #2 (Recommendations for the authors):*

There is a lot to like here including the elegant in vivo model system for cTag-PAPERCLIP in mouse cortical neurons with mTORC1 hyperactivation and multiple relevant in vitro systems to support and extend the findings. But the reader is left wondering what the impact of Trim9 APA has on cellular physiology (Figure S2A?) and if phosphorylation of CPSF6 via TORC1 signalling alters its subcellular localisation as described in *Drosophila* by Tang et al. (2018). The discussion does not make much of this evolutionary conservation, something that could easily be done with reference to (or by combining) Figures S3 and S4G. Are any CPSF6-dependent APA in *Drosophila* also driven by twin UGUA motifs?

The connection between APA and metabolism is likely much more pervasive than currently understood so this work is an important addition to the field. However, in this context, it seems disingenuous to limit the discussion of the Jong et al. (2019) eGFP plasmids to phosphorylation-state when the S8YD-CPSF6 was shown to be defective in nuclear accumulation, while S8YA, like wild-type was mostly nuclear. To support the current title, i.e., the connection between mTORC1 signalling and CPSF6 function it is strongly recommended that Figure 4I is extended to include the inhibitors (as in Figure 3G) or TORC1 hyperactivation (siTSC2). Based on Tang et al. (2018) knock down (or chemical inhibition) of CDK8 would result in a shift to cytoplasmic CPSF6. Ideally, microscopy of such experiments would confirm localisation data from Jong et al. (2019) and connect the research hear to the *Drosophila* data and evolutionarily conserved process. It would also clear up a current discrepancy in Figure 4I where WT and S8YD appear to restore relative PAS usage whereas S8YA does not, a finding that is hard to reconcile based on their reported localisation (see below).

I ask that the comments not be interpreted as negative, but rather from the intention to inform where/how I struggled to understand what was going on. If any comments are made in error, this should be taken as indication that as currently presented the data/text are unclear.

Specific comments on the manuscript that would help with clarity:

– It is important that in the first paragraph of page 9, you state that Tsc1-KO causes a shift toward Trim9-L mRNA use as this is assumed later rather than actually said.

– When written as Trim9/TRIM9-S (or Trim9/TRIM9-L), it makes it ambiguous whether you are referring to total mRNA/short form or short form/short form so it may be better to write this as Trim9-S/TRIM9-S. This is the same for Trim9/TRIM9-L. Throughout the text the nomenclature gets complicated as to what is mRNA, protein, human and mouse, splicing APA and UTR-APA TRIM9 and BRD4 vs BMPR1B and MOB4. It might be helpful find more explicit nomenclature.

– For description of Figure 3F, it should be stated that unlike in mice, siCPSF6 significantly increased TRIM9-L usage in human as based on what is said it would be easy to assume the results are the same in mouse and human.

– At the top of page 12, should be "These results suggest that…" (typo there).

– At the bottom of page 12, should be "we compared L3-WT usage between wildtype…" (remove in).

– At the top of page 16, should be "we found that the BRD4-L-to-BRD4-S ratio" (add hyphens).

– At the bottom of page 19, should be "not distinct to Trim9/TRIM9" (remove hyphen).

Comments on Figures:

In general, the data is well presented but not always consistent with respect to how equivalent data are presented (e.g. relative expression of TRIM9 isoforms vs ratio of isoform expression), labelling or colour scheme. It is recommended that the final version of the manuscript addresses these discrepancies. Detailed commentary for Figures1-4 follows but is not limited to those figures.

– 1A. Adding a timeline (eg. 2-weeks) would help.

– Figure 1B. The Trim9 transcript has been flipped in its orientation indicating a data manipulation that is not reflected in the legend.

– Figure 1C. In the accompanying text it would help the reader to understand the functional significance of losing/gaining SPRY by APA.

– Figure 1E and F. Unclear why there's a box around the heading. It seems inconsistent with other figures. These are also the first where it's not always clear what 'relative expression' on the y-axis means. If the intention is to set up the difference between absolute and ratio data it would be better to make the labelling of the y-axis in the ratio part of 1F consistent with those in Figure 2. Albeit why, data in Figure 2 are expressed as ratio rather than relative amounts (as similar data are expressed in Figure 3) is unclear.

– For Figure S2A, it is unclear how the statement "in Torin 1-treated N2a cells, Trim9 expression was consistently shifted toward Trim9-S with similar magnitudes at both mRNA and protein levels." The data appears to relate to confirmation of specific knock down of either both the long and short or long-only Trim9 isoform by siRNA. If this was used as a control for quantitation, or if a phenotype was monitored for these cells it should be noted.

– Overall in Figure 2 it is unclear what, if anything the colours mean. This is a recurrent issue, where the reader expects there to be a logic, such as coding of mRNA type, cell type etc… but is left wondering what they are missing.

– Figure 3G. Expressed as a ratio, it is difficult to understand if an altered ratio is due to loss of the short isoform (eg Figure 3B and 3D) or increase of the long form (Figure 3F). Moreover, a stickler for statistics might wish to know how the compound error is calculated.

– Figure 4I. The data here should be expressed consistent with 4F if the reader is to be able to interpret the data. Including the data from 4F would help to understand if the relative PAS usage is a recovery to normal or over-shoots that. As indicated above, it is hard to understand how S8YD-CPSF6 can function like WT-CPSF6 when according to Jang et al. (2019), S8YD-CPSF6 is cytosolic. At a minimum this needs to be explained in the text.

*Reviewer #3 (Recommendations for the authors):*

As a non-expert in mouse genetics or molecular cloning, I found the initial explanation of the generation of Tsc1-KO mouse models for PAPERCLIP analysis to be difficult to follow. To make the paper accessible to a wider audience greater explanation of terms and methods such as "floxing" and the AAV-PHP.eB system would be desirable.

Testing of TRIM9, BMPR1B and MOB4 reporters where each of two UGUA motifs in the twin motif is individually mutated would be necessary to convincingly show that the full UGUAYUGUA motif is required for CFIm regulation of these genes – and that this is a generally important cis-regulatory element in alternative polyadenylation.

The description of the bioinformatic analysis in the methods section is incomplete and refers to previously published methods instead of giving details. *eLife* guidelines suggest that authors should provide "complete information… to ensure that readers can easily understand what was measured and analysed". The CIMS package that was used for some of the analysis is not cited and is difficult to find (may have been superseded by another package called CTK). Publications for Cutadapt and Novoalign are also not cited. It is also not clear what statistical method was used to identify genes with alternative polyadenylation in the mutant condition. Please expand upon this section of the methods.

The authors rely heavily on bar charts with error bars to visualise their qPCR and western blotting results. *eLife* statistical reporting guidelines recommend that raw data should be presented in figures when the number of replicates is less than 10. This could be done using stripplots. For qPCR data, the figures also present unlogged data transformed relative to control conditions, which skews the data and makes error bars difficult to interpret. I would suggest instead presenting the data using -dCt values instead.

---

## [Author Response]

Essential revisions:Reviewer #1 (Recommendations for the authors):1. The initial descriptions of the genetic challenges of the breeding schemes of the mice used is almost impenetrable. Much of this could be shifted to the methods. I appreciate the authors attention to technical detail, but momentum is immediately lost in this paper without much gain in technical resolution. The authors should consider consolidating the language.

Reviewer 3 also raised this issue. We condensed the 3 paragraphs in the section into 1 paragraph and moved some information into the Methods section.

2. The description of the two criteria for selecting candidate APA events for further investigation included 'resulted in expression of two different protein isoforms'. The authors need to clarify and justify this criteria. Only ~10% of APA events are splicing-APA in that distinct C-termini are created. By implementing this criteria, the authors artificially trimmed the list down by a huge margin. I think this is still ok, but more explanation is required so that the average reader understands what went into this filtering event.

We expanded the paragraph (page 8) to explain the two types of APA shifts and the rationale to implement the filtering strategy.

3. Can the authors describe what a SPRY domain is for non-experts?

We added a new sentence with citations (page 8) to describe the SPRY domain as suggested:

“SPRY domain is commonly found in TRIM family proteins and is implicated in protein-protein interaction.”

We also added the full name of SPRY domain to the Figure 1C legend (page 25).

4. The section of the results focused on cdk8 or CLK2 is under-developed. What doe cdk8 or CLK2 inhibitors do in the absence of Torin1? Can the authors show that cdk8 inhibition does anything to phosphorylation of CPSF6? As it stands, this section is not well-controlled or analyzed and needs improvement.

Please see below.

5. Continuing along the lines of the previous point, CPSF6 phosphorylation is also under-developed. Where are these phospho-mutants localized? Do cdk8 manipulations also effect these reporter experiments? While the UGUA mutagenesis experiments are well done, the authors could consider removing the phosphorylation experiments altogether and not have their overall thesis impacted at all.

We agree that the CDK8/CLK2 and CPSF6 phosphorylation sections could be expanded and substantiated. As our new experiments strengthened the role of CFIm in *Trim9* regulation by mTORC1, we decided to remove both sections as suggested.

Reviewer #2 (Recommendations for the authors):There is a lot to like here including the elegant in vivo model system for cTag-PAPERCLIP in mouse cortical neurons with mTORC1 hyperactivation and multiple relevant in vitro systems to support and extend the findings. But the reader is left wondering what the impact of Trim9 APA has on cellular physiology (Figure S2A?) and if phosphorylation of CPSF6 via TORC1 signalling alters its subcellular localisation as described in *Drosophila* by Tang et al. (2018). The discussion does not make much of this evolutionary conservation, something that could easily be done with reference to (or by combining) Figures S3 and S4G. Are any CPSF6-dependent APA in *Drosophila* also driven by twin UGUA motifs?The connection between APA and metabolism is likely much more pervasive than currently understood so this work is an important addition to the field. However, in this context, it seems disingenuous to limit the discussion of the Jong et al. (2019) eGFP plasmids to phosphorylation-state when the S8YD-CPSF6 was shown to be defective in nuclear accumulation, while S8YA, like wild-type was mostly nuclear. To support the current title, i.e., the connection between mTORC1 signalling and CPSF6 function it is strongly recommended that Figure 4I is extended to include the inhibitors (as in Figure 3G) or TORC1 hyperactivation (siTSC2). Based on Tang et al. (2018) knock down (or chemical inhibition) of CDK8 would result in a shift to cytoplasmic CPSF6. Ideally, microscopy of such experiments would confirm localisation data from Jong et al. (2019) and connect the research hear to the *Drosophila* data and evolutionarily conserved process. It would also clear up a current discrepancy in Figure 4I where WT and S8YD appear to restore relative PAS usage whereas S8YA does not, a finding that is hard to reconcile based on their reported localisation (see below).I ask that the comments not be interpreted as negative, but rather from the intention to inform where/how I struggled to understand what was going on. If any comments are made in error, this should be taken as indication that as currently presented the data/text are unclear.

The original figures S3 and S4G are now combined and presented in Figure 7—figure supplement 1C. In addition, we created a new Figure 7—figure supplement 1D to show the lack of twin UGUA motif in *Drosophila Atg1* and *Atg8a* PASs. We also revised the first paragraph of Discussion (page 20) to incorporate the evolution aspects as suggested.

Specific comments on the manuscript that would help with clarity:– It is important that in the first paragraph of page 9, you state that Tsc1-KO causes a shift toward Trim9-L mRNA use as this is assumed later rather than actually said.

A new sentence, “*Tsc1* knockout in mouse cortical excitatory neurons causes a shift toward *Trim9-L* mRNA expression (Figure 1B)”, was added as suggested (page 8, second paragraph).

– When written as Trim9/TRIM9-S (or Trim9/TRIM9-L), it makes it ambiguous whether you are referring to total mRNA/short form or short form/short form so it may be better to write this as Trim9-S/TRIM9-S. This is the same for Trim9/TRIM9-L. Throughout the text the nomenclature gets complicated as to what is mRNA, protein, human and mouse, splicing APA and UTR-APA TRIM9 and BRD4 vs BMPR1B and MOB4. It might be helpful find more explicit nomenclature.

We change all Trim9/TRIM9-S and Trim9/TRIM9-L into Trim9-S/TRIM9-S and Trim9-L/TRIM9-L to improve the clarity as suggested.

– For description of Figure 3F, it should be stated that unlike in mice, siCPSF6 significantly increased TRIM9-L usage in human as based on what is said it would be easy to assume the results are the same in mouse and human.– At the top of page 12, should be "These results suggest that…" (typo there).– At the bottom of page 12, should be "we compared L3-WT usage between wildtype…" (remove in).– At the top of page 16, should be "we found that the BRD4-L-to-BRD4-S ratio" (add hyphens).– At the bottom of page 19, should be "not distinct to Trim9/TRIM9" (remove hyphen).

All 4 listed typos/errors are either removed or corrected.

Comments on Figures:In general, the data is well presented but not always consistent with respect to how equivalent data are presented (e.g. relative expression of TRIM9 isoforms vs ratio of isoform expression), labelling or colour scheme. It is recommended that the final version of the manuscript addresses these discrepancies. Detailed commentary for Figures1-4 follows but is not limited to those figures.– 1A. Adding a timeline (eg. 2-weeks) would help.

A timeline (“2~3 weeks”) is added to Figure 1A as suggested.

– Figure 1B. The Trim9 transcript has been flipped in its orientation indicating a data manipulation that is not reflected in the legend.

A new sentence, “(Because *Trim9* is located on the minus strand, the orientation is horizontally flipped from the original.)”, was added to the Figure 1B legend (page 25) as suggested.

– Figure 1C. In the accompanying text it would help the reader to understand the functional significance of losing/gaining SPRY by APA.

We added a new sentence with citations (page 8, second paragraph) to describe the SPRY domain as suggested by Reviewer 1:

“SPRY domain is commonly found in TRIM family proteins and is implicated in protein-protein interaction”.

We also added the full name of SPRY domain to the Figure 1C legend (page 25).

– Figure 1E and F. Unclear why there's a box around the heading. It seems inconsistent with other figures. These are also the first where it's not always clear what 'relative expression' on the y-axis means. If the intention is to set up the difference between absolute and ratio data it would be better to make the labelling of the y-axis in the ratio part of 1F consistent with those in Figure 2. Albeit why, data in Figure 2 are expressed as ratio rather than relative amounts (as similar data are expressed in Figure 3) is unclear.

1) Both boxes around the heading are removed from Figure 1EandF.

2) We thoroughly revised all bar graphs showing *Trim9* isoform shifts and expanded some of the figure legends to improve clarity. In N2a cells, *Cpsf6* or *Nudt21* loss-of-function had stronger effects on individual *Trim9* isoforms compared to the effects from changes in mTORC1 activities. Nevertheless, changes in mTORC1 activities did consistently alter the relative expression between the two *Trim9* isoforms in both directions. Therefore, the original Figure 2 was constructed to emphasize the consistent shift. In the revised manuscript, for all experiments, the expression of individual isoform is also shown either in the main or supplemental figures in addition to the relative expression between isoforms.

– For Figure S2A, it is unclear how the statement "in Torin 1-treated N2a cells, Trim9 expression was consistently shifted toward Trim9-S with similar magnitudes at both mRNA and protein levels." The data appears to relate to confirmation of specific knock down of either both the long and short or long-only Trim9 isoform by siRNA. If this was used as a control for quantitation, or if a phenotype was monitored for these cells it should be noted.

A new sentence,

“We also performed siRNA experiments in N2a cells to characterize a TRIM9 antibody (Qin et al., 2016) for *Trim9-S* and *Trim9-L* protein detection (Figure 2—figure supplement 1A)”,

was added (page 10, first paragraph) to avoid confusion.

– Overall in Figure 2 it is unclear what, if anything the colours mean. This is a recurrent issue, where the reader expects there to be a logic, such as coding of mRNA type, cell type etc… but is left wondering what they are missing.

We are sorry about the confusion caused by all the different colors. We thoroughly revised all figures with a unifying color scheme: For endogenous genes (*TRIM9* or *BRD4*), individual isoform expression is colored in different shades of blue while the relative expression between the two isoforms is colored in orange. For reporter assays, different colors are used to represent distinct reporters.

– Figure 3G. Expressed as a ratio, it is difficult to understand if an altered ratio is due to loss of the short isoform (eg Figure 3B and 3D) or increase of the long form (Figure 3F). Moreover, a stickler for statistics might wish to know how the compound error is calculated.

We decided to remove the CDK8/CLK2 and CPSF6 phosphorylation sections as suggested by Reviewer 1 and the original Figure 3G was removed as a result. Nevertheless, we felt it important to clarify the language/label used to express the relative expression between two *Trim9* isoforms below.

In the revised manuscript, we reserve the Y-axis label of “TRIM9-L/TRIM9-S RNA ratio” specifically for the right panel of Figure 1F, in which ‘absolute’ expression of each *Trim9* isoforms can be obtained from RNA-seq measurement. For all other figures, in which RT-qPCR was used for measurement and quantification was calculated using the ΔΔCt method (no ratio was calculated), we use labels like “TRIM9-L RNA expression (relative to TRIM9-S)” to highlight the difference. We expanded the Figure 2 legend (page 25~27) to describe the plotting process and set up for the subsequent similar figures.

– Figure 4I. The data here should be expressed consistent with 4F if the reader is to be able to interpret the data. Including the data from 4F would help to understand if the relative PAS usage is a recovery to normal or over-shoots that. As indicated above, it is hard to understand how S8YD-CPSF6 can function like WT-CPSF6 when according to Jang et al. (2019), S8YD-CPSF6 is cytosolic. At a minimum this needs to be explained in the text.

We decided to remove the CDK8/CLK2 and CPSF6 phosphorylation sections and the original Figure 4I was removed as a result.

Reviewer #3 (Recommendations for the authors):As a non-expert in mouse genetics or molecular cloning, I found the initial explanation of the generation of Tsc1-KO mouse models for PAPERCLIP analysis to be difficult to follow. To make the paper accessible to a wider audience greater explanation of terms and methods such as "floxing" and the AAV-PHP.eB system would be desirable.

Reviewer 1 also raised this issue. We condensed the 3 paragraphs in the section into 1 paragraph and moved some information into the Methods section.

Testing of TRIM9, BMPR1B and MOB4 reporters where each of two UGUA motifs in the twin motif is individually mutated would be necessary to convincingly show that the full UGUAYUGUA motif is required for CFIm regulation of these genes – and that this is a generally important cis-regulatory element in alternative polyadenylation.

We generated new TRIM9-S PAS, BMPR1B distal PAS, and MOB4 distal PAS reporters and performed reporter assays in 293T cells to examine how mutations in individual UGUAs of the twin UGUA motif affect the PAS usage. The results are presented in the new Figure 4H-I and 5I-K, which demonstrate that the effects from individual UGUAs of the twin UGUA motif are not additive in all 3 PASs. It is also clear that neither of the individual UGUA is solely responsible for the full effects from the entire twin UGUA motif in TRIM9-S PAS and BMPR1B distal PAS*.* Overall, the new results support that it might be advantageous to have a second copy of UGUA and they were incorporated into the Discussion section (page 21, second paragraph).

The description of the bioinformatic analysis in the methods section is incomplete and refers to previously published methods instead of giving details. eLife guidelines suggest that authors should provide "complete information… to ensure that readers can easily understand what was measured and analysed". The CIMS package that was used for some of the analysis is not cited and is difficult to find (may have been superseded by another package called CTK). Publications for Cutadapt and Novoalign are also not cited. It is also not clear what statistical method was used to identify genes with alternative polyadenylation in the mutant condition. Please expand upon this section of the methods.

The APA shift analysis paragraph was expanded to include descriptions from Hwang et al. 2016 (page 36, 3^rd^ paragraph) and the citations for the CIMS package, Cutadapt and Novoalign were added as requested (page 36, second paragraph)*.*

The authors rely heavily on bar charts with error bars to visualise their qPCR and western blotting results. eLife statistical reporting guidelines recommend that raw data should be presented in figures when the number of replicates is less than 10. This could be done using stripplots. For qPCR data, the figures also present unlogged data transformed relative to control conditions, which skews the data and makes error bars difficult to interpret. I would suggest instead presenting the data using -dCt values instead.

We revised all bar graphs to show individual data points with a unified color scheme*.*